# Physics-Informed Conditional Diffusion for Multi-Modal PDEs

## Abstract

Many physical systems that are represented by partial differential equations (PDEs) admit multiple valid solutions, such as eigenstates of differential operators, or wave modes, yet most neural PDE surrogates are deterministic and collapse to averages. This multiplicity of solutions is especially predominant in various engineering and scientific domains ranging from acoustics and seismology to quantum systems. With the ability to generate or complete sparse measurements, diffusion-based approaches to solve PDEs by sampling physically valid solutions are gaining traction as an alternative to traditional numerical solvers. In this paper, we present a novel physics-informed conditional diffusion framework for multi-modal PDEs, called PDEDIFF, that learns distributions over solution fields from sparse, irregular samples while enforcing governing equations and boundary conditions through mesh-free residual penalties computed by automatic differentiation. PDEDIFF is capable of effectively solving PDEs with multiple valid solutions by learning $\mathbb{P}[Y|X]$, i.e., it learns a solution field $Y$ for a corresponding input spatial information $X$. Unlike Physics-Informed Neural Networks (PINNs), which minimize residuals around expected values $\mathbb{E}[Y|X]$ and hence tend to regress toward a conditional mean, PDEDIFF samples diverse physically consistent solutions by integrating PDE residuals directly into the diffusion objective. Our results indicate that generative, physics-informed diffusion is a practical tool for uncertainty-aware and multi-modal PDE modeling in low-to-moderate dimensions.

## 1 Introduction

Accurate modeling and solving of partial differential equations (PDEs) is fundamental to advancing scientific disciplines, areas ranging from acoustics and seismology to quantum systems and fluid dynamics. Physics-Informed Neural Networks (PINNs) Raissi et al. (2019a) have emerged as a powerful approach for embedding known physical laws into machine learning models. In recent years, PINNs have excelled in incorporating domain-specific equations and information into the learning process by adding residual terms, which include differential equation and boundary conditions, to a regressor's loss, allowing data-efficient training from limited observations. Despite their ability to leverage domain knowledge and efficiently estimate conditional means ($\mathbb{E}[Y|X]$), PINNs collapse multi-modal target distributions to a single mean-field solution and hence blur distinct physically admissible solutions or eigenstates and fail to capture these multiple scenarios.

Diffusion models, driven by advances in deep generative modeling, have achieved remarkable performance in tasks requiring detailed sampling from complex probability distributions such as in Ho et al. (2020b); Song et al. (2020). In this paper, we propose using this capability of reversing a noise process to sample from solution fields.

Diffusion models have further advanced into multiple domains such as image synthesis Rombach et al. (2022); Kawar et al. (2022); Kakinuma et al. (2025), healthcare Cao et al. (2024); Chung & Ye (2022), and drug discovery Alakhdar et al. (2024); Corso et al. (2023). Conditional variants can, in principle, capture the distribution over solution fields $\mathbb{P}[Y|X]$ rather than a single point estimate, offering critical advantages in problems where multiple solutions naturally arise such as parameterized PDEs or in eigen-PDEs. However, standard diffusion models typically do not incorporate physical constraints explicitly, resulting in samples that may violate conservation laws, boundary conditions, or any other operator constraints.

In consideration of the above problems with diffusion models, there are some recent works, like Bastek et al. (2025) and Shu et al. (2023), that include physics residual as penalties during training. Yet they targeted cases where the PDE's solution pairs can be sampled from a single mode distribution. Also, the above methods retained a grid-based input for training and generation that only support computation residuals using finite-difference stencils.

This ability to sample multiple modes is particularly advantageous in solving PDEs with multiple solutions, such as Schrodinger equation in quantum mechanics with multiple eigenfunctions Lundeen et al. (2011) or multiple wave solutions in Helmholtz equation. Another important setting where multiple solutions arise is in PDEs with unknown parameters or boundary/initial conditions. These unknown parameters can either be missing from data recorded during experiments or often be expensive to measure or reconstruct, thus requiring that solutions from multiple parameter settings be identified that fit the observed measurements. For example, an experimental dataset may contain records of acoustic waves using multiple sensors in a room, however the source locations (initial conditions) or room configuration (boundary conditions) for each experiment may not be fully recorded. In this case, solving the Helmholtz equation to sample multiple solutions that conform to the observed sensor readings, while accounting for the physics, would be of interest. In contrast, standard approaches such as finite difference solvers or even PINNs may either guess a few parameter values and present solutions for them, or learn a single mean estimated parameter value across all experiments.

Motivated by these challenges, we introduce PDEDIFF, a mesh-free, physics-informed conditional diffusion framework that learns distributions over solution fields from sparse, irregular samples while steering generation toward physically valid solutions. The model conditions on coordinates and uses automatic differentiation to evaluate PDE residuals and boundary or initial conditions directly at sampled points, therefore no fixed mesh or finite-difference stencils are required. PDEDIFF's interface is flexible and compatible with different PDEs and we only replace the residual and boundary conditions. Therefore, this makes PDEDIFF a complement to classical solvers as these methods are mesh-specific and equation-specific, whereas our method can be trained once and then work as a generative surrogate that can produce physics aware solutions at new coordinates without the need for re-meshing.

The key contributions of our work include the following:

- Generative solver for multi-modal PDEs: PDEDIFF learns distributions over solution fields and samples distinct, physically admissible modes instead of regressing to an average.

- Physics-conditioned sampler: A coordinate-conditioned encoder–decoder (CCED) denoiser that embeds both spatial coordinates and diffusion time, trained with a dual loss on data fidelity and residual physics.

- Mesh-free residual enforcement: Higher-order derivatives are computed via autograd, enabling irregular or sparse point clouds rather than fixed grids as used in previous work on physics-informed diffusion models.

- Comprehensive evaluations: We benchmark our method on Infinite Potential Wells, Gross-Pitaevskii Equation, and Helmholtz Equation, reporting the Wasserstein-1 distance and modality recall against analytical ground truth. We have also experimented on simpler toy problems with circles similar to the ones in Bastek et al. (2025). Our results indicate that PDEDIFF outperforms PINNs on PDEs with multiple solutions.

## 2 RELATED WORK

### 2.1 DIFFUSION MODELS

With the development of diffusion models Song et al. (2020),Ho et al. (2020b), and Song et al. (2021), their applications for solving PDEs are emerging. Most of them involve a conditioned diffusion using score-based method to guide the solution based on sparse observations. Huang et al. (2024b) proposed training procedure and sampling strategies that are conditioned on incoming observations to recover accurate PDEs trajectories based on the score-based approach Song et al. (2021). Qu et al. (2024) consider the inverse problem of data assimilation where they aim to recover the complete weather state from observations of various modalities. Rühling Cachay et al. (2023) proposed a fully data-driven framework using temporal interpolation as the forward process and forecasting as the reverse diffusion process for spatial-temporal forecasting problems. However,

relying solely on data-driven approaches may fail to capture the underlying physical laws, potentially leading to samples that lack physical consistency or generalizability beyond the training data. Another diffusion-based work Du et al. (2024) focus on chaotic systems, but their approach is also completely data-driven and implicitly learns the physics distribution with high-fidelity samples.

## 2.2 PHYSICS-INFORMED MODELS

PINNs Raissi et al. (2017) have appeared as a robust framework for solving both forward and inverse problems governed by PDEs. The key idea behind PINNs is to incorporate the PDE directly into the loss function of a neural network. This is achieved by penalizing the network's predictions when they deviate from the physical laws represented by the PDE. As a result, PINNs can learn solutions even from sparse or noisy training data while ensuring that the learned function respects the underlying physics. The line of work by Jin et al. (2022) uses feed-forward neural network to learn the multiple eigenfunctions for the family of Schrödinger's equations. They achieved multiple solutions through penalizing with several physics-informed regularizers and iteratively training to learn new eigenfunctions. However, this approach is only applicable to eigenproblems and would not be generalized to other types of PDE. Recently, some attention-based methods such as Wu et al. (2024) use physics-aware attention to learn a deterministic solver on dense meshes and irregular geometries, but the learned attention patterns remain unconstrained and may violate fundamental physical laws such as boundary conditions, and limiting their reliability.

Towards physics informed diffusion-based approaches, CocoGen Jacobsen et al. (2024) and DiffusionPDE Huang et al. (2024a) inject the governing equations into the sampling process of score-based generative models to enforce the consistency of the samples with the underlying PDE, but physics guidance was only applied to the last $N$ steps. They also focus on the reconstruction problem that conditioned on grids of sparse measurement. Shysheya et al. (2024) focused on forecasting tasks with conditioning training and sampling via the score-based approach. Recent works by Bastek et al. (2025) and Shu et al. (2023) use the Denoising Diffusion Probabilistic Model (DDPM) architecture combined with physics-informed loss during training. The physics-informed guidance in Shu et al. (2023) is applied with some probability $p_t$ (a hyperparameter) as an additional input to the UNET architecture in DDPM. Bastek et al. (2025) models the physics term as a virtual likelihood and combines this with the diffusion training loss. Above all, these works employed the finite difference method to compute the physics-informed guidance. This limited the generalization of the model as we need to construct the finite difference scheme for every new PDE, which can become complicated and unstable for high-order derivatives. Also, computing a finite difference required a fixed grid for consistency. Our work makes use of the automatic differentiation technique to circumvent these issues.

## 3 BACKGROUND

### 3.1 PARTIAL DIFFERENTIAL EQUATION CONSTRAINTS

We consider the following general form for a system of PDEs defined on some domain $\Omega$

$$\begin{cases} \mathcal{F}(u(x)) = f(x), & x \in \Omega \\ \mathcal{B}(u(x)) = b(x), & x \in \partial\Omega \end{cases} \tag{1}$$

where $\mathcal{F}$ is the interior differential operator for instance, Laplacian or Hamiltonian, $\mathcal{B}$ encodes the boundary condition, $\partial\Omega$ is the boundary of domain $\Omega$, and $u$ is the solution field that satisfies the set of PDEs for all $x \in \Omega$ and boundary conditions $x \in \partial\Omega$. $f$ and $b$ are functions independent of $u$, usually representing external conditions applied to the system. In physics-informed learning setup, along with the data loss, which is the $l_2$ loss between the generated samples and ground truth samples, we also aim to minimize a physics residual $R$. This physics residual is calculated by taking $l_2$ difference measuring the extent to which boundary and interior physics constraints are satisfied at model prediction $\hat{u}$ as given in Raissi et al. (2019b).

$$R(\hat{u}(x), x) = \lambda_{int}\mathcal{L}_\Omega + \lambda_{bc}\mathcal{L}_{bc} \tag{2}$$
$$= \lambda_{int}||\mathcal{F}(\hat{u}(x)) - f(x)||_2 + \lambda_{bc}||\mathcal{B}(\hat{u}(x)) - b(x)||_2 \tag{3}$$

here $\hat{u}(x)$ is the solution predicted by a deep learning model, and $\mathcal{L}_{bc}$ is the physics residual of the boundary conditions and $\mathcal{L}_\Omega$ is the physics residual of PDE constraints defined in the interior of the domain $\Omega$. $\lambda_{int}$ and $\lambda_{bc}$ are the weights that we want to consider for the respective residuals.

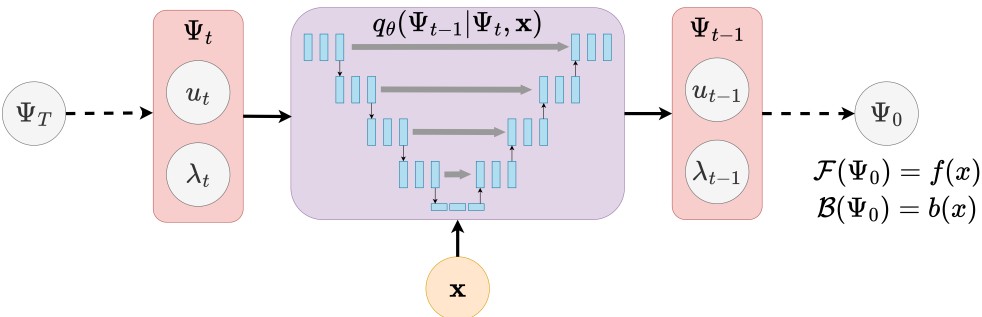

Figure 1: PDEDIFF denoising step denoises random noise into solution field and eigenvalues by conditioning on the spatial coordinates $\mathbf{x}$. The goal of this coordinate conditioned encoder decoder (CCED) architecture is to generate samples that follow the residual $\mathcal{F}$ and boundary conditions $\mathcal{B}$

For problems dealing with multiple solutions either through eigen-states or parameterized PDEs, we will be training the generative model to generate both the solution field and eigenvalues/parameters for the differential operators; we revisit this in detail in Section 4.1.

Usually when closed form of a differential operator does not exist, researchers generally opt for a finite difference approach to calculate the derivatives, but this method becomes very unstable for higher order derivatives and that is why we will be using automatic differentiation to eliminate this uncertainty in gradient calculation and making it more flexible than using a grid-based approach.

## 3.2 DENOISING DIFFUSION PROBABILISTIC MODEL

Diffusion models are a class of deep generative models inspired by non-equilibrium thermodynamics. Ho et al. (2020b) models the forward process by iteratively adding noise to the data so that it resembles a random distribution. Formally, Eq. 4 defines the forward diffusion process, here $u_0$ is the true data and noise sampled from a gaussian distribution is added over time steps $t \in 1, \ldots, T$:

$$p(u_t|u_{t-1}) = \mathcal{N}(u_t; \sqrt{1 - \beta_t}u_{t-1}, \beta_t I), \tag{4}$$

here $\{\beta_t\}_{t=1}^T$ controls the variance schedule in the above equation and after large enough $T$ steps, $u_T$ will resemble a sample from a gaussian distribution. The objective of a diffusion model is to learn a reverse (denoising) process that denoises a sample generated from $\mathcal{N}(0, I)$ into samples from the data distribution $p_{\text{data}}$. Mathematically, the model and the reverse process parameterized by model parameters $\theta$ is defined as:

$$q_\theta(u_{t-1}|u_t) = \mathcal{N}(u_{t-1}; \mu_\theta(u_t, t), \Sigma_\theta(u_t, t)), \tag{5}$$

This allows the generation of samples from complex data distributions. In the case of conditional diffusion models, additional information (such as class labels or prompts) is used to condition the denoising network. The model then approximates the full conditional density and generates outputs sampled from the distribution of a desired output class or prompt.

## 4 METHOD

In this section, we will dive deep into PDEDIFF, a novel conditioned diffusion model framework that learns a distribution corresponding to solution field and parameters in a PDE agnostic setting. We will begin with formalizing the learning task (Sec. 4.1), and derive the physics-informed conditional diffusion algorithm (Sec. 4.1). We will then describe the architecture and the training objective (Sec. 4.2 and Sec. 4.3) and then introduce the metrics used to benchmark the results (Sec. 4.4).

## 4.1 PROBLEM FORMULATION

Similar to a standard data-driven approach, we will provide the independent variables as input to the diffusion model. The forward and reverse process of the diffusion model then outputs our dependent variables. For our problem setting, we will be working with different PDEs such as Time Independent Schrödinger's Equation (TISE), and non-homogeneous Helmholtz equation. The input for each of the corresponding PDE will be the spatial coordinates $x$ and the output is the corresponding solution field and the parameter that satisfies the PDE. In this paper, we will use $u(x)$ to denote

the solution field or eigenfunction and $\lambda$ to denote the parameter of the differential operator or the eigenvalue. Both terms will be used interchangeably as we discuss different PDEs.

Our method is designed to learn distributions over feasible solutions fields and eigenvalues of the PDE directly from data. For given spatial coordinates $x \in \Omega \subset \mathbb{R}^d$, the diffusion model is conditioned on $x$ to generate the dependent variables $u(x)$ and $\lambda$. The goal of the model is to learn to generate samples that follow the below PDE:

$$\mathcal{H}_\lambda u(x) = g(x) \tag{6}$$

here $\mathcal{H}_\lambda$ is a linear or non-linear differential operator (usually of second-order or higher order) dependent on parameter $\lambda$, and $g : \Omega \to \mathbb{R}$ is a known function, and $u : \Omega \to \mathbb{R}$ is the solution field. The role of $\lambda$ in different PDEs works as a system parameter, for instance, in the case of eigen-PDEs such as the Schrodinger equation, $\lambda$ represents the eigenvalue or the energy of the corresponding wavefunction. In the case of the non-homogeneous Helmholtz equation, $\lambda = k$ is being used to denote the wavenumber that could correspond to different frequencies and waves that have been collected from a physical setup. In general form of PDE, $\lambda$ will represent the physical system parameters such as conductivity, viscosity, Young's Modulus and many more depending on the physical system and the PDE describing the system. When PDEDIFF perform sampling, it samples the solution field and this system parameter jointly. For simplicity, we only consider PDE with Dirichlet boundary condition in our problem formulation, that is, $\mathcal{B}(u(x)) = 0$. But the method can be expand to more general setting of any multi-solution PDEs with non-zero boundary condition (Sec. A.5.5).

Given a distribution $p(x)$ where $x \in \Omega$, we are interested in learning the multi-modal conditional density function $p(u(x), \lambda \,|\, x)$ from the sparse dataset $\{(x^{(n)}, u^{(n)}, \lambda^{(n)})\}_{n=1}^N$, whose samples follow the PDEs. For simplicity, we will use $\Psi = (u, \lambda)$ to denote a data point, and $p(\Psi|x)$ to denote a probability distribution. The forward diffusion process is simulated by gradually adding controlled Gaussian noise to $\Psi$, which can be model as a conditional distribution $p(\Psi_{t+1}(x)|\Psi_t(x), x) \sim \mathcal{N}(\sqrt{1 - \beta_t}\Psi_t(x), \beta_t I)$, where $\{\beta_t\}_{t=1}^T$ is a sequence of noise scheduler.

$$\underbrace{\Psi_T|x}_{\text{Gaussian noise}} \leftarrow \Psi_{T-1}|x \leftarrow \cdots \leftarrow \Psi_t|x \leftarrow \cdots \leftarrow \Psi_1|x \leftarrow \underbrace{\Psi_0|x}_{\text{real data}}$$

This process eventually yields a structured latent representation at timestep $T$. We adapt the method from Ho et al. (2020b); Song et al. (2020) to derive a simplified loss function for the denoising process. Note that the reverse distribution $q(\Psi_t(x)|\Psi_{t+1}(x), x)$ is intractable, we therefore use a neural network $\text{NN}(\Psi_t, x, t)$ to model the inverse diffusion steps $q_\theta(\Psi_{t-1}|\Psi_t, \Psi_0, x)$ that maximize the log-likelihood of $q(\Psi_0|x)$:

$$\log q(\Psi_0|x) = \log \int q(\Psi_{0:T}|x) d\Psi_{1:T} \tag{7}$$

This log-likelihood is not tractable due to unknown denoising process. Therefore, we will approximate it with evidence lower bound, which is easier to optimize:

$$\log q(\Psi_0|x) \geq \mathbb{E}_{p(\Psi_{1:T}|\Psi_0, x)} \left[ \log \frac{q(\Psi_{0:T}|x)}{p(\Psi_{1:T}|\Psi_0, x)} \right] \tag{8}$$

Simplifying the evidence lower bound (further details in the Appendix A.1) gives us

$$\log q(\Psi_0|x) \geq \mathbb{E}_{p(\Psi_1|\Psi_0, x)} \left[ \log \frac{q_0(\Psi_0|\Psi_1, x)}{p(\Psi_1|\Psi_0, x)} \right]$$
$$+ \sum_{t=2}^T \underbrace{\mathbb{E}_{p(\Psi_t|\Psi_0, x)} \left[ D_{KL}(p(\Psi_{t-1}|\Psi_t, \Psi_0, x) || q_\theta(\Psi_{t-1}|\Psi_t, x)) \right]}_{= \mathcal{L}_{\text{data}, t}} \tag{9}$$

In addition to maximizing the likelihood of $q(\Psi_0|x)$, we are also interested in incorporating the physics guidance in training the diffusion model. Our goal is also to minimize the likelihood of the residual $q(R(\Psi_0, x)|\Psi_0, x) \sim \mathcal{N}(R(\Psi_0, x), \sigma^2)$, where the variance $\sigma^2$ approaches 0 as the model learns $\Psi_0$. In this paper, we consider PDEs of the form $\mathcal{H}_\lambda u(x) = g(x)$. Below is how this *PDE-loss* or $\mathcal{L}_{\text{residual}}$ is defined:

$$\log q(R(\Psi_0, x)|\Psi_0, x) \propto \|\mathcal{H}_\lambda \hat{u}_0 - g\|_2^2 =: \mathcal{L}_{\text{residual}}(\hat{\Psi}_0) \tag{10}$$

**Algorithm 1** PDEDIFF Training

1: **Input:** Training dataset $\Psi_0 = (u_0(x), \lambda_0)$
2: **Output:** Trained denoising process $\text{CCED}_\theta$
3: **repeat**
4:      $x \sim \text{Uniform}(\Omega)$
5:      $\Psi_0(x) \sim q(\Psi_0|x)$ (from training data)
6:      $t \sim \text{Uniform}\{1, \ldots, T\}$
7:      $\epsilon \sim \mathcal{N}(0, I)$
8:      $\bar{\alpha}_t = \prod_{s=1}^{t}(1 - \beta_t)$
9:      $\Psi_t(x) = \sqrt{\bar{\alpha}_t}\Psi_0(x) + \sqrt{1 - \bar{\alpha}_t}\,\epsilon$
10:     $\hat{\Psi}_0 = \text{CCED}_\theta(\Psi_t(x), x, t)$
11:     Compute $\nabla_\theta \mathcal{L}_{\text{total},t}(\hat{\Psi}_0)$ (as in 11)
12:     Update $\text{CCED}_\theta$ via GD
13: **until** converged

**Algorithm 2** PDEDIFF Sampling

1: **Input:** Number of samples $N$, trained denoising process $\text{CCED}_\theta$
2: **Output:** Set of samples $\{\Psi_0\}_{i=1}^{N}$
3: **for** $i = 1$ to $N$ **do**
4:      Sample $x_i \sim \text{Uniform}(\Omega)$
5:      **for** $t = 1$ to $T$ **do**
6:          $\hat{\Psi}_0(x_i) = \text{CCED}_\theta(\Psi_t(x_i), x_i, t)$
7:          Sample $\Psi_0(x_i)$ with DDIM Song et al. (2020)
8:      **end for**
9: **end for**

## 4.2 TRAINING AND SAMPLING ALGORITHMS

We follow a similar training objective and approach as given in Ho et al. (2020a); Bastek et al. (2025), where we add a physics informed loss as a regularizer to the entire loss function. The training algorithm in Bastek et al. (2025) uses a finite difference stencil method to calculate the derivatives of the quantities required in the residual. We instead leverage the power of automatic differentiation (autodiff) to calculate the gradients for the residual equations. This strategy provides more flexibility to calculate higher order derivatives as compared to using finite difference stencils, which become unstable when calculating higher order derivatives. Moreover, the finite difference method can only be feasible where the data points are arranged in a regular grid and all the points on the grid correspond to only a single solution, whereas PDEDIFF can train on data points that are randomly sampled from a multi-modal distribution and each generated sample can correspond to any of the feasible solutions.

We will now introduce the objective function of PDEDIFF that is a loss function composed of the standard reconstruction error and physics informed constraints. The loss function is defined as:

$$\mathcal{L}_{\text{total},t}(\hat{\Psi}_0) = \mathbb{E}_{t \sim [1:T], p(\Psi_{1:T}|x)} \left[ c_{\text{data}}\mathcal{L}_{\text{data},t}(\Psi_0, \hat{\Psi}_0) + c_{\text{res}}\mathcal{L}_{\text{residual},t}(\hat{\Psi}_0) \right] \tag{11}$$

where we use it to train a denoising process as in Algorithm 1.

Here $\mathcal{L}_{\text{data}}$ is called the data loss which is the reconstruction error between the predicted output and the ground truth values, and for our framework we have used a standard Mean Square Error (MSE) loss. $\mathcal{L}_{\text{residual}}$ is the residual loss, which penalizes the model when the generated samples do not satisfy the underlying physics constraints, i.e., the solutions generated must follow the PDEs. The terms $c_{data}$ and $c_{res}$ are hyperparameters that we set during training to weight the importance of the data loss and physics loss, respectively. Since, we provide the model with the initial spatial and temporal information, we can leverage the use of autodiff to calculate high order derivatives for calculating the residual. This approach also allows us to use randomly sampled data points instead of going with a grid-based approach providing us with more flexibility with respect to the dataset used and also for the sampling part for inference. For PDEDIFF sampling Algorithm 2, we utilize the sampling method from Denoising Diffusion Implicit Model (DDIM) Song et al. (2020) to accelerate the generation process. Also, in step 4 of Algorithm 2, we could replace uniform samples of $x_i$ with any other grid or mesh coordinates for applicable uses.

## 4.3 ARCHITECTURE

PDEDIFF employs a *coordinate conditioned encoder decoder (CCED)* denoiser that converts a noisy input into its corresponding denoised output by conditioning on the spatial coordinates (independent variable) and diffusion time-step. We pass the noisy input along with the spatial coordinates $x \in \Omega \subseteq \mathbb{R}^d$ and the diffusion time $t$ into the denoiser where each layer is called a *Conditional Block*, which consists of a set of linear and embedding layers. Figure 1 demonstrates how the denois-

| Setting | | Vanilla | | Physics-informed | |
|---|---|---|---|---|---|
| | | PMSE | WD | PMSE | WD |
| 1D (2 states) | P | 0.822 | 0.539 | 1.052 | 0.344 |
| | D | 0.932 | 0.157 | 0.935 | 0.157 |
| | N | 0.809 | 0.544 | 0.960 | 0.446 |
| | Q | 0.017 | 0.086 | **0.015** | **0.086** |
| 1D (3 states) | P | 0.371 | 0.737 | 0.460 | 0.442 |
| | D | 0.438 | 0.114 | 0.437 | 0.114 |
| | N | 0.333 | 0.776 | 0.445 | 0.470 |
| | Q | 0.022 | 0.055 | **0.012** | **0.050** |
| 1D GP (3 states) | P | 1.550 | 2.036 | 1.291 | 2.418 |
| | D | 0.868 | 1.430 | 0.867 | 1.431 |
| | N | 0.069 | 1.180 | **0.069** | 1.190 |
| | Q | 0.140 | 0.195 | 0.126 | **0.194** |

Table 1: Performance comparison between PINN (P), DeepONet Lu et al. (2021) (D), Physics-informed Neural Operator Li et al. (2021) (N) and PDEDIFF (Q) across different problem settings for 1D examples. PMSE here is P-MSE and WD is the Wasserstein metric. The lower values represent better match between the ground truth and the sampled wavefunctions. Details of the energy states will be provided in Appendix A.5.1. The first 2 rows correspond to 1D Infinite Potential Well, and 1D GP corresponds to 1D Gross-Pitaevskii Equation.

| Setting | | Vanilla | | Physics-informed | |
|---|---|---|---|---|---|
| | | PMSE | WD | PMSE | WD |
| 2D (3 states) | P | 1.253 | 0.378 | 1.239 | 0.340 |
| | D | 7.327 | 0.411 | 7.326 | 0.410 |
| | N | 1.254 | 0.297 | 1.398 | 0.502 |
| | Q | 1.222 | 0.169 | **1.194** | **0.156** |
| 2D(4 states) degenerate | P | 1.250 | 0.482 | 1.243 | 0.477 |
| | D | 7.661 | 0.485 | 7.659 | 0.485 |
| | N | 1.332 | 0.299 | **0.933** | 0.513 |
| | Q | 1.309 | 0.150 | 1.338 | **0.124** |
| 2D (4 states) non-degenerate | P | 1.133 | 0.597 | 1.123 | 0.623 |
| | D | 22.34 | 0.513 | 22.34 | 0.513 |
| | N | 1.198 | 0.344 | 1.257 | 0.587 |
| | Q | 1.100 | 0.222 | **1.091** | **0.205** |
| Helmholtz Equation | P | 0.800 | 0.107 | 0.746 | 0.108 |
| | D | 21.35 | 0.168 | 21.35 | 0.168 |
| | N | 0.753 | 0.111 | 1.000 | 0.106 |
| | Q | 0.253 | 0.092 | **0.227** | **0.091** |

Table 2: Performance comparison between PINN (P), DeepONet Lu et al. (2021) (D), Physics-informed Neural Operator Li et al. (2021) (N) and PDEDIFF (Q) across different problem settings. PMSE here is P-MSE and WD is the Wasserstein metric. The lower values represent better match between the ground truth and the sampled wavefunctions. Details of the energy states for 2D Schrodinger experiments will be provided in Appendix A.5.3, and of the frequency parameters for the non-homogeneous Helmholtz equation will be provided in Appendix A.5.4.

ing model converts random noise into the corresponding solutions functions and their corresponding eigenvalues or parameters. The details can be found in Appendix A.3.

Further, we use ADAMW for training and we also experiment with different weights for the residual loss ($c_{res}$) (additional results on this is discussed in Appendix A.8). For baselines, we compare our method with a vanilla conditional diffusion model ($c_{res} = 0$) with no physics regularization, and we also compare PDEDIFF with a physics-informed fully-connected neural network (PINN) Raissi et al. (2019b), physics-informed neural operator (PINO) Li et al. (2021) and DeepONet Lu et al. (2021). We use the same values of the physics-informed hyperparameter $c_{res}$ for all comparative studies and ablation results. Since for the training algorithm we do not require to implement a custom finite difference stencil and instead proceed with using autodiff, this makes our framework compatible with mesh-free datasets.

### 4.4 QUANTIFICATION METRICS

Given some spatial input $\mathbf{x} \in \Omega$, PDEDIFF would generate $\Psi_0 = (u_0, \lambda_0)$ corresponding to this $\mathbf{x}$. The PDEs that it is trying to solve has multiple correct solutions, for e.g., the first 3 wavefunctions and energy values for the Schrödinger's Equation could correspond to the following possible solutions $(u_0^{(1)}, \lambda_0^{(1)}), (u_0^{(2)}, \lambda_0^{(2)}), (u_0^{(3)}, \lambda_0^{(3)})$. Since each of these is a correct solution to the PDE, we need a metric that can effectively compare the generated samples with the true distribution or the ground truth solutions.

**P-MSE:** This metric calculates the mean squared error of the samples generated with the ground-truth eigenstates or solutions modes. Formally, the metric can be defined as below:

$$P\text{-MSE} = \mathbb{E}_{\mathbf{x} \sim \Omega} \left[ \left\| \left( \hat{u}_0(\mathbf{x}), \hat{\lambda}_0(\mathbf{x}) \right) - \left( u^{(k(\mathbf{x}))}, \lambda^{(k(\mathbf{x}))} \right) \right\|_2^2 \right] \quad (12)$$

$$\text{where} \quad k(\mathbf{x}) = \underset{i \in \{1, \dots, M\}}{\arg\min} \, ||\hat{\lambda}_0(\mathbf{x}) - \lambda^{(i)}||_2 \quad (13)$$

Here, $M$ is the number of feasible solution modes, $(\hat{u}_0(\mathbf{x}), \hat{\lambda}_0(\mathbf{x}))$ is the generated sample for a particular $\mathbf{x}$, and $k(\mathbf{x})$ is the function that assigns a particular sample to its closest solution field by comparing the parameter values and this state is then used to calculate the MSE.

**Earth-Mover's Distance (Wasserstein-1)**: For two probability measures $p, q$ on a domain $\Omega \subset \mathbb{R}^d$ we use the 1-Wasserstein distance (WD), here $\Gamma(p, q)$ denotes the set of couplings with marginals $p$ and $q$

$$W_1(p, q) = \inf_{\gamma \in \Gamma(p, q)} \int_{\Omega \times \Omega} \|\mathbf{x} - \mathbf{y}\| \, \mathrm{d}\gamma(\mathbf{x}, \mathbf{y}), \quad (14)$$

This metric is used to compare the distances between two probability distributions and in some PDEs, wavefunctions have a special property, i.e., it also represents the probability densities of a particle, i.e., $|\psi|^2$ represents the probability density function of the particle. The ground truth distribution is represented by a linear combination of the probability distribution of the individual solution modes, and the generated distribution can be calculated by squaring the predicted $\psi(\mathbf{x})$.

To compare the predicted probability distribution and the true distribution, we evaluate both the functions on the same set of point cloud $\mathbf{x}$. Then the distributions are normalized as below:

$$p_{\text{pred}}(\mathbf{x}) = \frac{|\psi_{\text{pred}}(\mathbf{x})|^2}{\sum_{\mathbf{x}' \in \mathcal{X}} |\psi_{\text{pred}}(\mathbf{x}')|^2} \qquad p_{\text{true}}(\mathbf{x}) = \sum_{i=1}^N \pi_i \frac{|\psi^{(i)}(\mathbf{x})|^2}{\sum_{\mathbf{x}' \in \mathcal{X}} |\psi^{(i)}(\mathbf{x}')|^2} \quad (15)$$

where $\sum_{i=1}^n \pi_i = 1$ (linear combination coefficients) and after calculating the probability distributions, we use the optimal transport library Flamary et al. (2021).

## 5 EXPERIMENT RESULTS

In this section, we present our experiment setup and results for several PDE problems with multiple solutions such as Time Independent Schrödinger Equation in 1D and 2D infinite potential well, non-homogeneous Helmholtz Equation, Gross–Pitaevskii Equation. More experiment results can be found in Sec. A.5 on some other equations like Burgers' Equation and toy examples on circle.

### 5.1 1D SCHRÖDINGER: PARTICLE IN A BOX

We consider the time-independent Schrödinger equation for a quantum particle in a one-dimensional infinite potential well:

$$\underbrace{\left( -\frac{\hbar^2}{2m} \partial_x^2 + V(x) - E \right)}_{\mathcal{H}_E} \psi(x) = 0, \quad V(x) = \begin{cases} 0, & x_L < x < x_R \\ \infty, & \text{otherwise} \end{cases}$$

where $V(x)$ denotes the infinite potential well, or in other words, the space of a 1D box that contains the particle where it moves freely inside but trapped between the wall $(x_L, x_R)$ defined as above. Note that $\partial_x^2$ are the second-order derivatives of $\psi$ with respect to $x$ and could be calculate using automatic differentiation to compute the physics informed loss during training. Given this PDE, we are trying to recover the solution field $\psi(x)$ and corresponding energy or parameter $E$. Results are presented in Table 1 and Figure 2.

### 5.2 1D SCHRÖDINGER: GROSS-PITAEVSKII EQUATION

We consider also one-dimensional time-independent Gross-Pitaevskii equation, a form of non-linear Schrodinger equation. In specific, the Halmiltonian operator $\mathcal{H}$ is non-linear and written as follow:

$$\underbrace{\left( \frac{h^2 \partial_x^2}{2m} + V(x) + U_0|\psi(x)|^2 - E \right)}_{\mathcal{H}_E} \psi(x) = 0$$

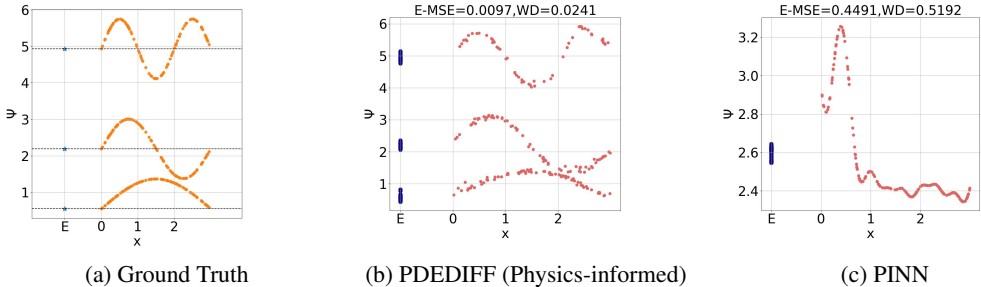

(a) Ground Truth     (b) PDEDIFF (Physics-informed)     (c) PINN

Figure 2: Comparing generated samples by PDEDIFF and PINN for 1D Schrödinger equation

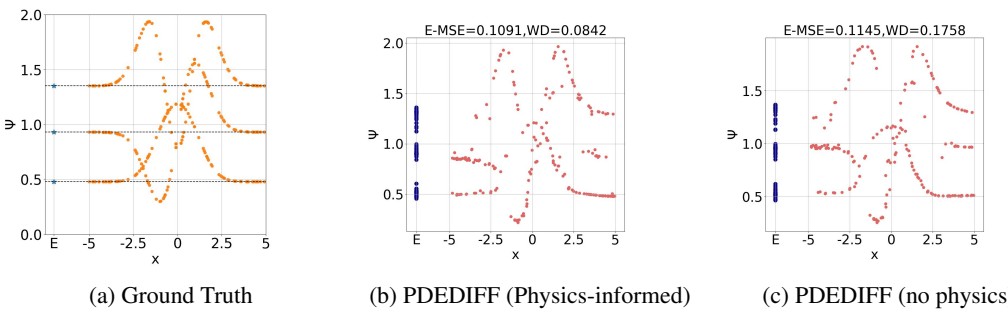

(a) Ground Truth     (b) PDEDIFF (Physics-informed)     (c) PDEDIFF (no physics)

Figure 3: Comparing generated samples by PDEDIFF for 1D Gross-Pitaevskii equation

where $U_0$ is interaction force, $V(x) = \frac{1}{2}mx^2$ is the harmonic trapping potential. Not many nonlinear Schodinger equations have closed-form solution, such as the one presented here. We obtain training samples and ground truth by solving the time-independent Gross-Pitaevskii equation numerically as in Chiofalo et al. (2000). Results are presented in Table 1, Figure 3 and 4.

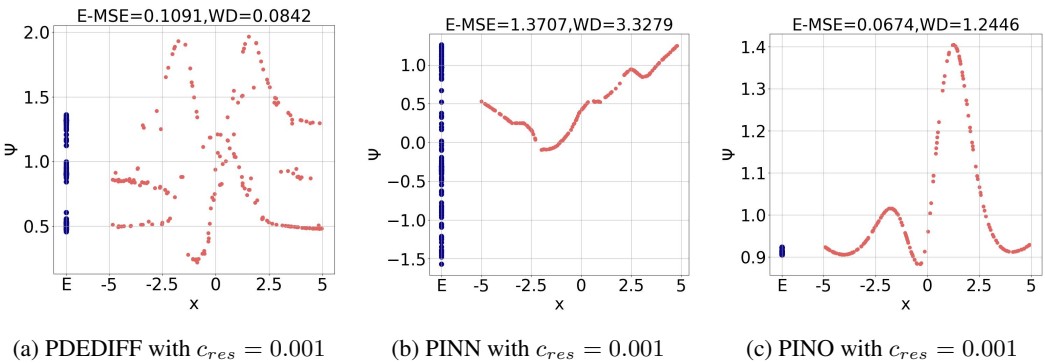

(a) PDEDIFF with $c_{res} = 0.001$     (b) PINN with $c_{res} = 0.001$     (c) PINO with $c_{res} = 0.001$

Figure 4: Generated samples by physics-informed PDEDIFF, PINN and PINO with $c_{res} = 0.001$ values for 1D Gross-Pitaevskii equation. Although PINO achieves a very low PMSE metric, it collapses to a single mode and results in a high WD metric.

### 5.3 2D SCHRÖDINGER: PARTICLE IN A BOX

We again consider the TISE for a quantum particle in a two-dimensional infinite potential well:

$$\underbrace{\left(-\frac{\hbar^2}{2m}(\partial_x^2 + \partial_y^2) + V(\mathbf{x}) - E\right)}_{\mathcal{H}_E} \psi(\mathbf{x}) = 0, \quad V(\mathbf{x}) = \begin{cases} 0, & x_L < x < x_R, \quad y_L < y < y_R \\ \infty, & \text{otherwise} \end{cases}$$

where $\mathbf{x} = (x, y)$ denotes the spatial coordinates and $V(\mathbf{x})$ denotes the 2D infinite potential well. $\partial_x^2$ and $\partial_y^2$ are the second-order partial derivatives of $\psi$ with respect to $x$ and $y$. The particle is confined inside the box $\mathbb{B} = [x_L, x_R] \times [y_L, y_R] \subset \mathbb{R}^2$ with Dirichlet boundary conditions, i.e., $\psi(\mathbf{x}) = 0$ for any $\mathbf{x} \in \partial\mathbb{B}$, where $\partial\mathbb{B}$ denotes the boundary of $\mathbb{B}$. The results are presented in Tab. 2.

## 5.4 HELMHOLTZ EQUATION

Another case study is the Helmholtz equation that arises from applications in heat conduction or acoustics. The Helmholtz equation is a non-homogeneous elliptic PDE express as follow:

$$\underbrace{\nabla^2 \psi(x) + k^2 \psi(x)}_{\mathcal{H}_k \psi} = f(x)$$

where integer or real valued parameter $k$ is known as frequency or wave number, $f$ is a source function that describe the emission source of the wave $\psi(x)$, with $x \in \Omega$. Our goal is to recover the high resolution propagation of the wave through the environment $\psi(x)$ and its frequency $k$. Training data is simulated by solving the second-order finite-difference linear system $\mathcal{H}_k \psi = f$ with low fidelity. The results are presented in Table 2.

## 6 DISCUSSION

We observed that PDEDIFF consistently achieves low MSE and WD metric, with physics-informed PDEDIFF achieving the best result in almost all cases (Tab. 1, 2) This demonstrates that PDEDIFF is capable of modeling, generating multimodal distributions and achieves higher numerical accuracy compared to the ground truth. As shown in Fig. 2, PDEDIFF effectively captures the multiple modes present in the training data, while standard feedforward neural networks tend to regress to the mean, failing to represent the distinct modes and instead producing oversmoothed approximations.

In the case of 1D Schrödinger equation, PDEDIFF has lower errors in both MSE and Wasserstein distance. Incorporating physics-informed guidance further improves performance. Nevertheless, Schrödinger equations have trivial solution where $\psi(x) = 0, E = 0$. This causes problems as with our physics guidance, since the physics loss is also minimized (equal to 0) if the model infers the trivial solution, as observed in our ablation study in A.8, 12 and 13. For the 1D Gross-Pitaevskii equation, PINO achieves a low P-MSE metric but this is due to convergence to a single mode, as observed in Fig. 4 and the high WD metric in Tab. 2, while PDEDIFF is able to obtain both low P-MSE and WD metric. We also demonstrate the capabilities of PDEDIFF on 2D Schrödinger equation. In the problem setting of 3 waves, the model is able to learn the different eigenstates and produce samples that are much closer to the true distribution and have a lower MSE loss. In the case where there are 4 waves, the model was trained on both setting wherestates where there are degenerate eigenstates, i.e., 2 eigenstates had the same energy but different wavefunctions, and non-degenerate eigenstates. PDEDIFF is yet able to distinguish the degenerate energy states and generate samples that have a lower Wasserstein metric.

In inhomogeneous Helmholtz Equation, we tested our framework with 2 parameters. In this setting, we observed that the model picks up the low frequency wave better and it sometimes missed the peak in high frequency wave. Nevertheless, after training for 5000 epochs, PDEDIFF achieves the lowest metric comparing to other baselines, which also trained with the same amount of training points and number of epochs (Tab. 2).

## 7 CONCLUSION

We introduced **PDEDIFF**, a physics-informed conditional diffusion framework that samples *entire ensembles* of solution fields and eigenvalues for multiple solution PDEs. In contrast, PINNs regress to a conditional mean or collapse to one of the modes. We have also demonstrated that our approach is more flexible than other approaches to incorporating physics in diffusion models in terms of the data samples being mesh-free and also for calculating the physics residuals using automatic differentiation. Also, unlike previous work, we provide the first evidence that physics informed diffusion models can provide better results than physics informed neural networks for PDEs with multiple solutions, and thus are more faithful to the physical law that they must follow. Our work shows a potential towards learning and generating new data for multi-modal PDEs.

Looking ahead, our framework can potentially be used to explore methods to discover unseen solution fields and eigenvalues that are not observed in data. Similar to recent works, such as Jin et al. (2022), extending PDEDIFF to learning solution fields in a data-free environment is an interesting direction. Scaling this framework to noisy and high-dimensional experimental datasets would enable developing real-time digital simulations of problems that could be studied in much detail and also accelerate the growth of drug discoveries and sustainable quantum technologies.

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

## A  APPENDIX

### A.1  DERIVATIONS OF SIMPLIFIED TRAINING LOSS

Continuing from Sec 4.1 of the main paper, we had briefly discussed the problem formulation of PDEDIFF. The inverse diffusion step for $q_\theta(\Psi_{t-1}|\Psi_t, \Psi_0, x)$ that maximizes the log-likelihood of $q(\Psi_0|x)$ is defined in Eq. 7 of the paper as:

$$\log q(\Psi_0|x) = \log \int q(\Psi_{0:T}|x)d\Psi_{1:T} \tag{16}$$

Below are the steps for deriving and simplifying the lower evidence bound for our conditioned DDPM. Note that $\log$ of an expectation over a distribution might be intractable, so we consider

optimizing the evidence lower bound like previous work Ho et al. (2020b); Song et al. (2020).

$$\log q(\Psi_0|x) = \log \int q(\Psi_{0:T}|x)d\Psi_{1:T}$$

$$= \log \int \frac{q(\Psi_{0:T}|x)}{p(\Psi_{1:T}|\Psi_0,x)} p(\Psi_{1:T}|\Psi_0,x)d\Psi_{1:T}$$

$$= \log \mathbb{E}_{p(\Psi_{1:T}|\Psi_0,x)} \left[ \frac{q(\Psi_{0:T}|x)}{p(\Psi_{1:T}|\Psi_0,x)} \right]$$

$$\geq \mathbb{E}_{p(\Psi_{1:T}|\Psi_0,x)} \left[ \log \frac{q(\Psi_{0:T}|x)}{p(\Psi_{1:T}|\Psi_0,x)} \right]$$

$$= \mathbb{E}_{p(\Psi_{1:T}|\Psi_0,x)} \left[ \log \frac{q(\Psi_T|\Psi_0,x)\prod_{t=1}^{T} q(\Psi_{t-1}|\Psi_t,x)}{\prod_{t=1}^{T} p(\Psi_t|\Psi_{t-1},\Psi_0,x)} \right]$$

$$= \mathbb{E}_{p(\Psi_{1:T}|\Psi_0,x)} \left[ \log \frac{q(\Psi_T|x)q_0(\Psi_0|\Psi_1,x)\prod_{t=1}^{T-1} q_\theta(\Psi_t|\Psi_{t+1},x)}{\prod_{t=}^{T} p(\Psi_t|\Psi_{t-1},\Psi_0,x)} \right]$$

$$= \mathbb{E}_{p(\Psi_{1:T}|\Psi_0,x)} \left[ \log \frac{q(\Psi_T|x)q_0(\Psi_0|\Psi_1,x)\prod_{t=2}^{T} q_\theta(\Psi_{t-1}|\Psi_t,x)}{\prod_{t=1}^{T} p(\Psi_t|\Psi_{t-1},\Psi_0,x)} \right]$$

Observed that products inside the log term could be rewritten as

$$\log q(\Psi_0|x) \geq \mathbb{E}_{p(\Psi_{1:T}|\Psi_0,x)} \left[ \log \frac{q(\Psi_T|x)q_0(\Psi_0|\Psi_1,x)}{p(\Psi_1|\Psi_0,x)} + \sum_{t=2}^{T} \log \frac{q_\theta(\Psi_{t-1}|\Psi_t,x)}{p(\Psi_t|\Psi_{t-1},\Psi_0,x)} \right]$$

$$= \mathbb{E}_{p(\Psi_{1:T}|\Psi_0,x)} \left[ \log \frac{q(\Psi_T|x)p_0(\Psi_0|\Psi_1,x)}{p(\Psi_1|\Psi_0,x)} \right]$$

$$+ \sum_{t=2}^{T} \mathbb{E}_{p(\Psi_{1:T}|\Psi_0,x)} \left[ \log \frac{q_\theta(\Psi_{t-1}|\Psi_t,x)}{p(\Psi_t|\Psi_{t-1},\Psi_0,x)} \right]$$

$$= \mathbb{E}_{p(\Psi_1|\Psi_0,x)} \left[ \log \frac{q_0(\Psi_0|\Psi_1,x)}{p(\Psi_1|\Psi_0,x)} \right] + \sum_{t=2}^{T} \mathbb{E}_{p(\Psi_{t-1},\Psi_t|\Psi_0,x)} \left[ \log \frac{q_\theta(\Psi_{t-1}|\Psi_t)}{p(\Psi_t|\Psi_{t-1},\Psi_0,x)} \right]$$

$$= \mathbb{E}_{p(\Psi_1|\Psi_0,x)} \left[ \log \frac{q_0(\Psi_0|\Psi_1,x)}{p(\Psi_1|\Psi_0,x)} \right]$$

$$+ \sum_{t=2}^{T} \underbrace{\mathbb{E}_{p(\Psi_t|\Psi_0,x)} \left[ D_{KL}(p(\Psi_{t-1}|\Psi_t,\Psi_0,x)||q_\theta(\Psi_{t-1}|\Psi_t,x)) \right]}_{=\mathcal{L}_{\text{data},t}}$$

## A.2 DETAILS OF THE STEPS FOR TRAINING AND SAMPLING ALGORITHM

We are also providing details of the training and sampling algorithm here for completeness. The forward diffusion process is simulated by gradually adding controlled Gaussian noise to $\Psi$, which can be model as a conditional distribution $p(\Psi_{t+1}(x)|\Psi_t(x),x) \sim \mathcal{N}(\sqrt{1-\beta_t}\Psi_t(x), \beta_t I)$, where $\{\beta_t\}_{t=1}^{T}$ is a sequence of noise scheduler.

### A.2.1 LATENT STEPS FOR TRAINING

We can use reparametrization trick to write $p(\Psi_{t+1}(x)|\Psi_t(x),x)$ as

$$\Psi_{t+1}(x) = \sqrt{1-\beta_t}\Psi_t(x) + \sqrt{\beta_t}\epsilon_t, \quad \epsilon_t \sim \mathcal{N}(0,I)$$

We can unroll $\Psi_{t+1}$ based on $\Psi_{t-1}$

$$\Psi_{t+1}(x) = \sqrt{1-\beta_t}(\sqrt{1-\beta_{t-1}}\Psi_{t-1}(x) + \sqrt{\beta_{t-1}}\epsilon_{t-1}) + \sqrt{\beta_t}\epsilon_t, \quad \epsilon_{t-1} \sim \mathcal{N}(0,I)$$

$$= \sqrt{1-\beta_t}\sqrt{1-\beta_{t-1}}\Psi_{t-1}(x) + \sqrt{1-\beta_t}\sqrt{\beta_{t-1}}\epsilon_{t-1} + \sqrt{\beta_t}\epsilon_t$$

$$= \sqrt{1-\beta_t}\sqrt{1-\beta_{t-1}}\Psi_{t-1}(x) + \sqrt{1-(1-\beta_t)(1-\beta_{t-1})}\epsilon$$

where $\epsilon \sim \mathcal{N}(0, I)$; and the third equal sign holds since sum of two Gaussian distribution is also a Gaussian distribution.

Unroll with respect to $t$, we have $\Psi_{t+1}$ in term of $\Psi_0$

$$\Psi_{t+1}(x) = \sqrt{1-\beta_t}\sqrt{1-\beta_{t-1}}\Psi_{t-1}(x) + \sqrt{1-(1-\beta_t)(1-\beta_{t-1})}\epsilon$$

$$= \prod_{s=1}^{t+1}\sqrt{1-\beta_s}\Psi_0(x) + \sqrt{1-\prod_{s=1}^{t+1}(1-\beta_s)}\epsilon$$

So for brevity of notation, we denote

$$\bar{\alpha}_{t+1} = \prod_{s=1}^{t+1}(1-\beta_s) \tag{17}$$

as used in Algorithm 1 of the main paper.

### A.2.2 DDIM SAMPLING

We adapt the sampling technique from Song et al. (2020) for more efficient sampling. Details are provided here for completeness.

In step 6 of Algorithm 2, we obtain

$$\hat{\Psi}_0(x_i) = \text{CCED}_\theta(\Psi_t(x_i), x_i, t) \tag{18}$$

We can update denoise step $\Psi_{t-1}(x_i)$ using the predicted $\hat{\Psi}_0(x_i)$ and $\Psi_t(x_i)$

$$\Psi_{t-1}(x_i) = \sqrt{\bar{\alpha}_{t-1}}\hat{\Psi}_0(x_i) + \sqrt{\frac{1-\bar{\alpha}_{t-1}}{1-\bar{\alpha}_t}}\left(\Psi_t(x_i) - \sqrt{\bar{\alpha}_t}\hat{\Psi}_0(x_i)\right) \tag{19}$$

with $\bar{\alpha}_t$ defined as in A.2.1.

### A.3 ARCHITECTURE IMPLEMENTATION

In Sec. 4.3 of the main paper, we had briefly discussed that PDEDIFF employs a coordinate-conditioned encoder decoder (CCED) denoiser that converts a noisy input into its corresponding denoised output by conditioning on the spatial coordinates.

The first step in the CCED forward pass is the encoder, where the noisy input, $\mathbf{x}$ and $t$ are passed through this encoder layer that has multiple sets of the *Conditional Block* and between each of these blocks, the activation function used is *softplus* or *tanh* based on the problem setting and the residual equations involved. This encoder converts all this data into a low-dimensional latent representation.

The next step is the bottleneck layer that also consists of the *Conditional Block* and captures the encoding within this low dimensional latent space to capture critical information required for denoising the input.

After the bottleneck layer, the latent information passes through the decoder, similar to the encoder, transforms the latent information into the original domain to give us the predicted noise. Every layer of this decoder also has a skip connection from the encoder layer to effectively relay the encodings to improve efficiency and accuracy.

We implement PDEDIFF in `PyTorch 2.5.1` with `CUDA 12.0` on an NVIDIA A100 GPU. All random seeds were fixed using `torch.manual_seed`, `np.random.seed`, and `torch.cuda.manual_seed`.

We use a lightweight CCED to predict $(\hat{\psi}, \hat{E})$ given noisy input $\mathbf{z}$, the diffusion timestep $t \in \{0, \ldots, T-1\}$, and the spatial coordinate $\mathbf{x}$:

Diffusion Hyperparameters:

- *Steps*: $T = 100$.
- *Noise schedule*: cosine, $\beta_t \in [10^{-5}, 10^{-2}]$

- *Objective*: DDPM $L_{\text{simple}}$ on $x_0$ plus physics residual weight $c_{\text{res}}$
- *Optimizer*: AdamW, lr $= 5 \times 10^{-4}$
- *Training*: 1000 updates, batch 32
- *Sampling*: deterministic DDIM "single-step"

The implementation of our code can be found at: code

### A.4 METRICS FOR TOY EXAMPLES

We discussed the P-MSE and Wasserstein metric for Schrodinger equations. For toy examples where we consider settings involve circles, the metrics are defined slightly different from that of the ones described in Section 4.4 of the main paper since we do not have explicit eigenvalues for circle setups, and distributions on circles are not similar to those belongs to wavefunctions. We used mean squared error (MSE) to and Wasserstein distance on uniform distribution of circles (setting dependent) to evaluate the generated samples against ground truth values.

#### A.4.1 MEAN SQUARED ERROR

This metric calculates the mean squared error of the samples generated with the ground truth circles coordinates. We compute the squared distance to all possible $y$-coordinates for an spatial $x$ and simply choose the minimum. Formally, the metric can be defined as below:

$$\text{MSE} = \mathbb{E}_{\mathbf{x} \sim \Omega} \left[ \min_{y_i \in \mathcal{Y}} \|\hat{y} - y_i\|_2^2 \right]$$

where $\quad \mathcal{Y}$ is set of possible solution of $y$-coordinates

#### A.4.2 WASSERSTEIN DISTANCE ON UNIFORM DISTRIBUTIONS

For two probability measures $p, q$ on a domain $\Omega \subset \mathbb{R}^d$ Denotes $\Gamma(p, q)$ the set of couplings with marginals $p$ and $q$

$$W_1(p, q) = \inf_{\gamma \in \Gamma(p,q)} \int_{\Omega \times \Omega} \| \mathbf{x} - \mathbf{y} \| \, \mathrm{d}\gamma(\mathbf{x}, \mathbf{y}), \tag{20}$$

where $\Gamma(p, q)$ denotes the set of couplings with marginals $p$ and $q$.

Let $\{x_i\}_{i=1}^n \subset \mathbb{R}$ denote sampled $x$-coordinates and let $\{\hat{y}_i\}_{i=1}^n$ be the predicted $y$-values from a model. We define a uniform distribution on $(x_i, \hat{y}_i)$ pairs. We define the predicted distribution to be

$$p_{\text{pred}}((x_i, \hat{y}_i)) = \frac{1}{n} \delta_{(x_i, \hat{y}_i)}$$

**Unit circle.** We define the distribution for ground truth pairs $(x_i, y_i)$ for each $x_i$ uniformly sampled from $[-1, 1]$ as follow

$$p_{\text{true}}((x_i, y_i)) = \frac{1}{2n} \delta_{(x_i, y_i)}, \quad \text{where } \delta_{(x_i, y_i)} = \begin{cases} 1, & \text{if } x_i^2 + y_i^2 = 1 \\ 0, & \text{otherwise} \end{cases}$$

**Disjoint circles** We define the distribution for ground truth pairs $(x_i, y_i)$ for each $x_i$ uniformly sampled from $[-1, 3]$ as follow

$$p_{\text{true}}((x_i, y_i)) = \sum_{k=1}^{2} \pi_k \frac{1}{2n} \delta_{(x_i, y_i)}$$

where

$$\delta_{(x_i, y_i)} = \begin{cases} 1, & \text{if } x_i^2 + y_i^2 = 1, x_i < 1 \text{ or } (x_i - 2)^2 + (y_i - 2)^2 = 1, x_i \geq 1 \\ 0, & \text{otherwise} \end{cases}$$

and $\pi_k$ is the weight if $(x_i, y_i)$ belongs to circle $k$ ($k = 1$ for circle centers at $(0, 0)$ and $k = 2$ for circle centers at $(2, 2)$)

**Concentric circles** We define the distribution for ground truth pairs $(x_i, y_i)$ for each $x_i$ uniformly sampled from $[-1, 1]$ as follow

$$p_{\text{true}}((x_i, y_i)) = \sum_{k=1}^{2} \pi_k \frac{1}{2kn} \delta_{(x_i, y_i)}$$

where

$$\delta_{(x_i, y_i)} = \begin{cases} 1, & \text{if } (x_i, y_i) \in \mathcal{C} \\ 0, & \text{otherwise} \end{cases}$$

with

$$\mathcal{C} = \mathcal{C}_1 \cup \mathcal{C}_2$$
$$\text{with } \mathcal{C}_1 = \{(x_i, y_i) : x_i^2 + y_i^2 = 9, \text{ and } x_i \notin [-1, 1]\}$$
$$\mathcal{C}_2 = \{(x_i, y_i) : x_i^2 + y_i^2 = 1 \text{ or } x_i^2 + y_i^2 = 9, \text{ and } x_i \in [-1, 1]\}$$

and $\pi_k$ is the weight if $(x_i, y_i)$ belongs to $\mathcal{C}_k$.

### A.5 EXPERIMENT SETUP

In this section, we continue from Section 5 of the main paper and discuss the results of PDEDIFF in the problem settings defined earlier. We first discuss more details of the problem settings, i.e. how did we generate data for training and hyperparameters used in each physics setting.

#### A.5.1 1DSCHRODINGER: INFINITE POTENTIAL WELL/ PARTICLE IN A BOX

We consider the time-independent Schrödinger equation for a particle in a one-dimensional infinite potential well:

$$E\psi(x) = \underbrace{\left(-\frac{\hbar^2}{2m}\partial_x^2 + V(x)\right)}_{H} \psi(x)$$

where $V(x)$ denotes the infinite potential well, or in other words, the space of a 1D box that contains the particle where it moves freely inside but trapped between the wall $(x_L, x_R)$. The potential well can be defined as

$$V(x) = \begin{cases} 0, & x_L < x < x_R \\ \infty, & \text{otherwise} \end{cases}$$

In this setting, the particle is confined within $(x_L, x_R)$, with Dirichlet boundary conditions $\psi(x_L) = \psi(x_R) = 0$.

Solving for $\psi(x)$, we can do it analytically as outside of $x_L, x_R$, $\psi(x) = 0$, so the wavefunction takes the form:

$$\psi(x) = \sqrt{\frac{2}{L}} \sin\left(\frac{n\pi x}{L}\right)$$

where $n$ are positive integer values.

Dataset Generation: The dataset for 1D infinite potential well was generated by uniformly sampling spatial coordinates between $[0, L_x]$. After sampling the spatial coordinates, we analytically calculate the wavefunction and the energy values. For our experiments, we simplify the constant values as $\hbar = 1, m = 1$, and the size of the well is taken as $L_x = 3$. We generate 100 data points for each of the first few eigenstates for training, i.e. we choose $n = 1, 2, 3$ for the 1D TISE 3 waves setting, and $n = 1, 2$ for the 1D TISE 2 waves setting.

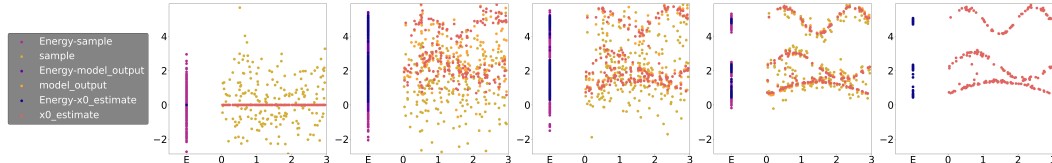

Figure 5: PDEDIFF denoising steps with $c_{res} = 0.001$ (left to right: denoise step $t = 100, 70, 40, 10, 0$). The left-most subplot shows noisy samples from Gaussian and each of the steps to the right shows the denoising process. The right-most subplot show the generated samples $\Psi_0$.

### A.5.2 1D GROSS-PITAEVSKII EQUATION

We consider also one-dimensional time-independent Gross-Pitaevskii equation, a form of non-linear Schrodinger equation. In specific, the Halmiltonian operator $\mathcal{H}$ is non-linear and written as follow:

$$E\psi(x) = \underbrace{\left( \frac{h^2 \partial_x^2}{2m} + V(x) + U_0 |\psi(x)|^2 \right) \psi(x)}_{\mathcal{H}}$$

where $U_0$ is interaction force, $V(x) = \frac{1}{2}mx^2$ is the harmonic trapping potential.

Not many non-linear Schodinger equations have closed-form solution, such as the one presented here. We obtain training samples and ground truth by solving the time-independent Gross-Pitaevskii equation via an explicit imaginary-time algorithm as in Chiofalo et al. (2000).

**Setup:** In this setting, the particle is free to move within $(x_L, x_R) \subset \mathbb{R}$, with Dirichlet boundary conditions $\psi(x_L) = \psi(x_R) = 0$. We generate 100 data points for each wavefunction and their corresponding energy using the algorithm in Chiofalo et al. (2000) for training and generate 200 points to evaluate our framework against the ground truth.

### A.5.3 2DSCHRODINGER: SQUARE INFINITE POTENTIAL WELL

We next consider the time-independent Schrödinger equation for a particle in a two-dimensional square infinite potential well:

$$E\psi(\mathbf{x}) = \underbrace{\left( -\frac{\hbar^2}{2m}(\partial_x^2 + \partial_y^2) + V(\mathbf{x}) \right) \psi(\mathbf{x})}_{H}$$

where $V(\mathbf{x})$ denotes the infinite potential well, or in other words, the space of a 2D box that contains the particle where it moves freely inside but trapped between the walls $\mathbb{B} = [x_L, x_R] \times [y_L, y_R] \subset \mathbb{R}^2$. The potential well can be defined as:

$$V(\mathbf{x}) = \begin{cases} 0, & x_L < x < x_R, \quad y_L < y < y_R \\ \infty, & \text{otherwise} \end{cases}$$

In this setting, the particle is confined within $\mathbb{B}$, with Dirichlet boundary conditions, i.e., $\psi(\mathbf{x}) = 0$ for any $\mathbf{x} \in \partial\mathbb{B}$, where $\partial\mathbb{B}$ denotes the boundary of the box $\mathbb{B}$.

Solving for $\psi(\mathbf{x})$, we can do it analytically as outside of $\mathbb{B}$, $\psi(\mathbf{x}) = 0$, so the wavefunction takes the form:

$$\psi(\mathbf{x}) = \frac{2}{L} \sin\left( \frac{n_x \pi x}{L} \right) \sin\left( \frac{n_y \pi y}{L} \right)$$

where $n_x, n_y$ are positive integer values.

**Set up**: The dataset for 2D infinite potential well was generated by uniformly sampling spatial coordinates between $x = [0, L_x], y = [0, L_y]$. After sampling the spatial coordinates, we analytically

calculate the wavefunction and the energy values. For our experiments, we simplify the constant values as $\hbar = 1, m = 1$, and the size of the well is taken as $L_x = L_y = 3$. For each eigenstate, we sample 200 data points for all the states in total. For 3 wave setting, we choose the states $(n_x, n_y)$ as $(1,1), (2,1), (2,2)$, and for the 4 wave settings, we experiment on degenerate states $(1,1), (1,2), (2,1), (2,2)$, i.e. some wavefunctions has the same energy, and nondegenerate states $(1,1), (2,1), (3,1), (4,1)$, i.e. every wavefunctions has distinct energy.

### A.5.4 HELMHOLTZ EQUATION

The Helmholtz equation is a non-homogeneous elliptic PDE that usually arises from applications in heat conduction, ultrasound or acoustics Basu & Rani (2021). It can be express as follow:

$$\underbrace{\nabla^2 \psi(x) + k^2 \psi(x)}_{\mathcal{H}_k \psi} = f(x)$$

where the integer or real valued parameter $k$ is known as frequency or wave number, and the source function $f$ describes the emission source of the wave $\psi(x)$, with $x = (x_1, x_2) \in \Omega \subseteq \mathbb{R}^2$. In our experiment, we use the following source function

$$f(x) = (1 + 8\pi^2) \cos(\pi^2 x_1) cos(\pi^2 x_2)$$

Our goal is to recover the high resolution propagation of the wave through the environment $\psi(x)$ and its frequency $k$. Training data is simulated by solving the second-order finite-difference linear system $\mathcal{H}\psi = f$ with low fidelity on a $16 \times 16$ grid. The results are presented in Tab. 2 and 11.

**Setup:** We sample 128 data points for each wave mode from a $16 \times 16$ grid and their corresponding frequency $k \in \{\pi, 4\pi\}$. The ground truth solution grids for training and evaluating our framework against the ground truth are obtained using the linear algebra solver on the discrete differential operator by second-order finite differences. Since only ground truth on fixed grid can be obtain through numerical method, we sample from fixed grid points for evaluation as in figure 6c, comparing against the ground truth in figure 6a and the prediction via PINN in figure 6b. In practical use cases, we can sample from any possible partial coordinate to complete the solution space via sampling from the learned PDEDIFF model, and hence can obtain multiple higher resolution solution fields on $32 \times 32$ grid with this method such as in figure 6d. The accuracy of the generated sample can be observed from slices of the 2D solution as in figures 6e-f for better visualization.

### A.5.5 BURGERS' EQUATION

We consider the 1D viscous Burgers' equation, a nonlinear PDE widely used as a simplified model of Navier-Stokes equations. In a more practical setting, we consider measurements of fluid dynamics with various viscosity. Assume that measurements of the fluid flow and viscosity are difficult and only low-fidelity, sparse measurements are obtained, and some governed physics are known in the form of the viscous Burger's equation Takamoto et al. (2023)

$$\partial_t u(t, x) + \frac{1}{2} \partial_x u^2(t, x) = \nu \partial_{xx} u(x, t), \quad x \in (0, 1), t \in (0, 2]$$

$$u(0, x) = u_0(x), \quad x \in (0, 1)$$

where $u(t, x)$ denotes the flow velocity and $\nu$ is the viscosity controlling smoothness of the solution and a periodic boundary condition.

**Setup:** We utilize the 1D Burgers' Equation dataset from PDEBench Takamoto et al. (2023). To simulate a scenario where governed physics are known but measurements are sparse, we isolate a fixed initial condition trajectory and observe its evolution under varying physical parameters. We extract solution fields for $\nu \in \{0.001, 0.4\}$ and subsample from the original high-resolution data. For training, we generate a dataset by uniformly sampling 100 spatial coordinate points for each (viscosity, time) pair, resulting in a sparse representation of the fluid flow across the spatiotemporal domain.

### A.5.6 TOY EXAMPLES

In this section, we discuss in more details for the experiment setup for the toy examples with circles. Given some $x$ coordinates that are uniformly sampled from a given range (such as $[-1, 1]$ for unit circle). We want to generate the corresponding $y$ coordinates that satisfy physics guidance.

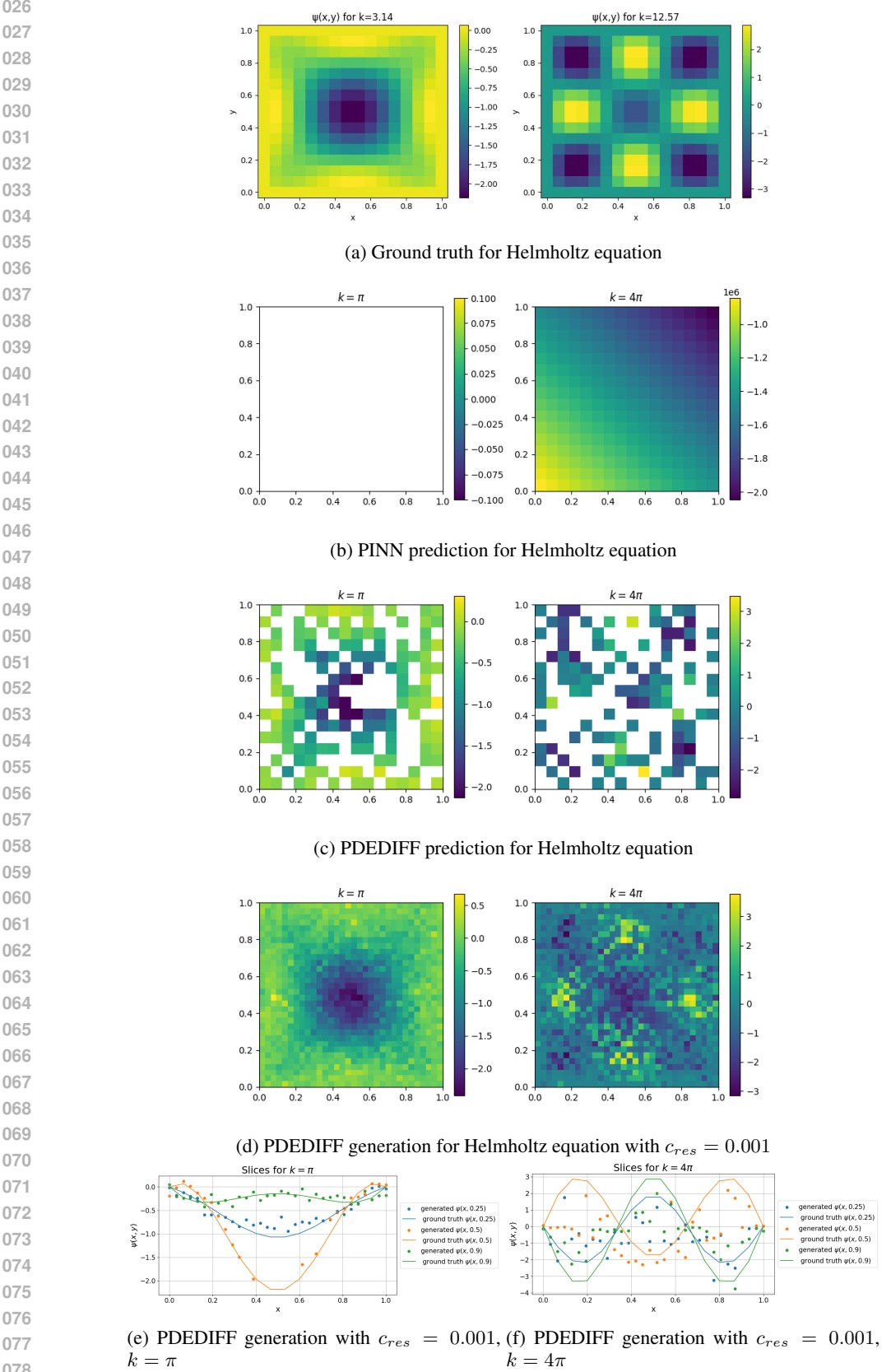

(a) Ground truth for Helmholtz equation

(b) PINN prediction for Helmholtz equation

(c) PDEDIFF prediction for Helmholtz equation

(d) PDEDIFF generation for Helmholtz equation with $c_{res} = 0.001$

(e) PDEDIFF generation with $c_{res} = 0.001$, (f) PDEDIFF generation with $c_{res} = 0.001$, $k = \pi$ $k = 4\pi$

Figure 6: Ground truth, prediction by PINN and generated samples by physics-informed PDEDIFF

| Setting | Burgers Equation | |
|---------|-----------------|---|
| | P | Q |
| 0 | $0.0516 \pm 0.0024$ | $0.0149 \pm 0.0025$ |
| 0.001 | $0.0685 \pm 0.0119$ | $\mathbf{0.0148 \pm 0.0023}$ |
| 0.005 | $0.0956 \pm 0.0019$ | $0.0199 \pm 0.0026$ |
| 0.01 | $0.1043 \pm 0.0045$ | $0.0233 \pm 0.0039$ |
| 0.05 | $0.1200 \pm 0.0009$ | $0.0375 \pm 0.0084$ |
| 0.1 | $0.1220 \pm 0.0007$ | $0.0406 \pm 0.0079$ |
| 0.5 | $0.1283 \pm 0.0004$ | $0.0496 \pm 0.0102$ |
| 1 | $0.1301 \pm 0.0028$ | $0.0540 \pm 0.0106$ |
| 10 | $0.1264 \pm 0.0137$ | $0.0710 \pm 0.0101$ |
| 100 | $0.1063 \pm 0.0494$ | $0.0782 \pm 0.0122$ |

Table 3: Preliminary performance comparison between PINN (P) and PDEDIFF (Q) for Burger's Equation for a particular trajectory with 2 different viscocities over the entire space and time values on P-MSE metric. The lower values represent better match between the ground truth and the sampled datapoints.

**Unit circle.** This problem could be formulated as a simple problem $\sqrt{1 - x^2} = \pm y$, in which the "eigen-values" $\pm 1$ is not explicitly exposed to the model. We instead required the model to learn this and output $y|x$ from provided training pairs $(x, y)$ and the physical constraints $x^2 + y^2 = 1$. Note that for a given $x$, there are 2 possible values for $y-$coordinates. We generate 100 points on the unit circle with coordinates $(x, y)$ satisfying $x^2 + y^2 = 1$ for training.

**Disjoint circle.** To study if the diffusion model is able to learn distinct distributions, we generate 200 points, with 100 points on a unit circle with coordinates $(x, y)$ satisfying $x^2 + y^2 = 1$ with centroid at $(0, 0)$, and another 100 points on a circle with coordinates $(x, y)$ satisfying $(x-2)^2 + (y-2)^2 = 1$ with center at $(2, 2)$. The two circles do not overlap each other.

**Concentric circle.** We generate 200 points, with 100 points on a circle with coordinates $(x, y)$ satisfying $x^2 + y^2 = 1$, and another 100 on a circle with coordinates $(x, y)$ satisfying $x^2 + y^2 = 9$. The two circles share the same centroid at $(0, 0)$. This creates a more complicated setting where for some $x$ coordinates, there exist 4 possible solutions. Figure 9 illustrates that PDEDIFF manages to distinguish different distributions, while PINN collapsed to only learning the mean of the data.

For circle settings, we consider the weighting for physics informed loss $c_{res} = 0.01$. To obtain new samples for visualization and error quantification, we generate 200 points with Algorithm 2 of the main paper. The generation by physics-informed PDEDIFF with $c_{res} = 0.01$ can be visualize at figures 7b, 8b, 9b for unit circle, disjoint circles and concentric circles, respectively. In comparison, generation by PINN (also with $c_{res} = 0.01$) are shown in Figures 7c, 8c, 9c, for unit circle, disjoint circles and concentric circles, respectively. For each toy setting, we run 5 experiments for each models on different seeds and present the average and one standard deviation of the metrics obtained with each model in table 4. In general, the performance of PDEDIFF with physics-informed loss outperforms that of PINN with similar residual coefficient $c_{res}$, assessed using the P-MSE and the Wasserstein metric. The experiments with toy examples of circles verify the performance of our method and show potential of physics-informed diffusion in distinguishing multimode distribution.

### A.6 COMPARISON WITH GAUSSIAN PROCESSES

We have performed a small set of experiments to compare PDEDIFF with Gaussian Processes as in Rasmussen & Williams (2005). These methods discuss how the differential operator could be embedded into the gaussian prior ensuring that the samples generated satisfy the governing equations. Even though Gaussian Process model uncertainty and can be used to generate samples from

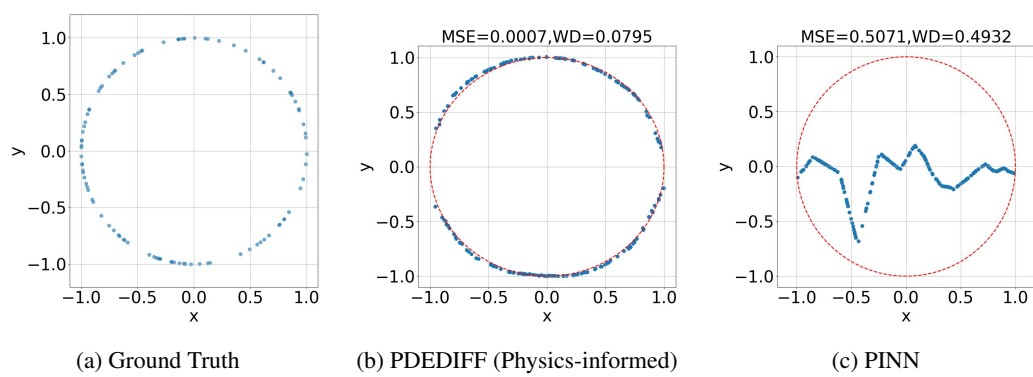

Figure 7: Comparison of generated samples by physics-informed PDEDIFF and PINN for unit circle. The dotted red line shows the ground truth and the blue dots are the inference point clouds.

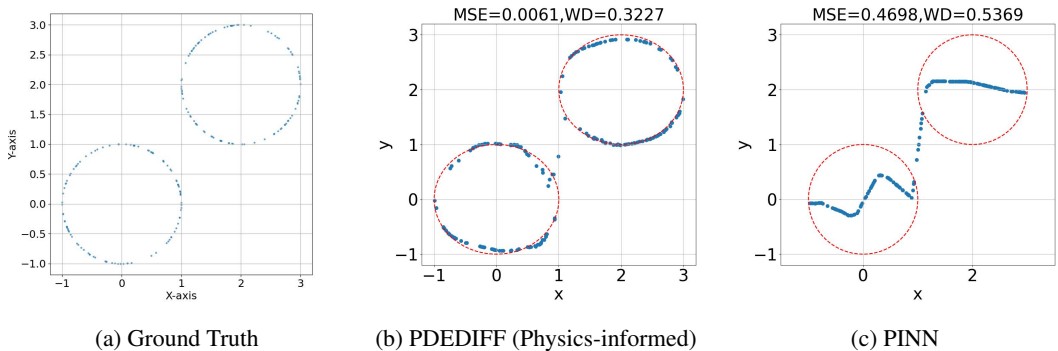

Figure 8: Comparison of generated samples by physics-informed PDEDIFF and PINN for disjoint circles. The dotted red line shows the ground truth and the blue dots are the inference point clouds.

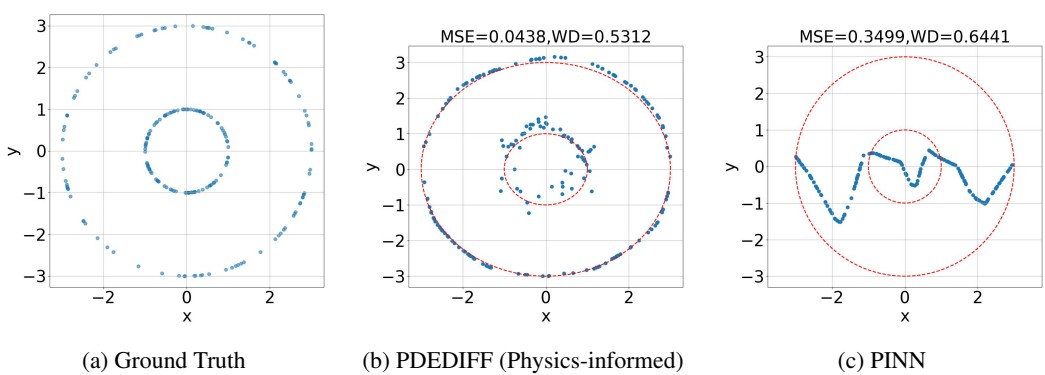

Figure 9: Comparison of generated samples by physics-informed PDEDIFF and PINN for concentric circles. The dotted red line shows the ground truth and the blue dots are the inference point clouds.

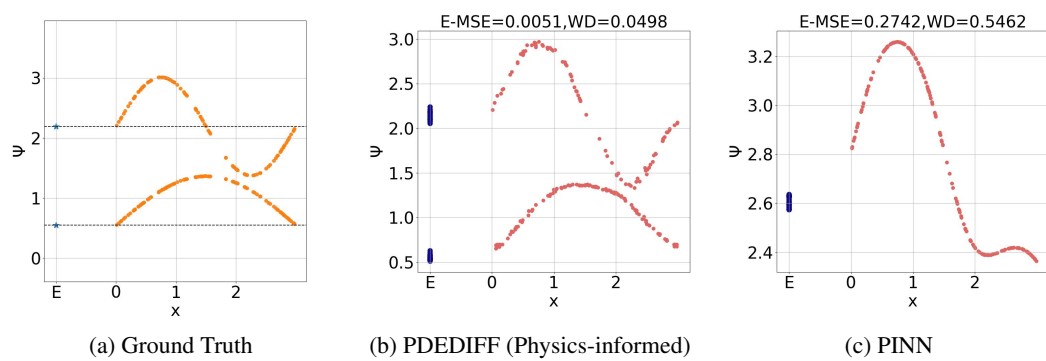

(a) Ground Truth      (b) PDEDIFF (Physics-informed)      (c) PINN

Figure 10: Comparison of generated samples by physics-informed PDEDIFF and PINN for 1D Schrödinger equation with 2 wavefunctions in infinite potential well.

the solution like diffusion models, the inherent Gaussian assumption implies a unimodal function, and hence the goal of this approach is still to learn the mean function. As a result, Gaussian Process based methods cannot handle settings where the PDEs have multiple feasible solutions.

We compare PDEDIFF with the EPGP method in Harkonen et al. (2023) on solving the 1D Schrodinger equation for infinite potential well with 2 states. For the Gaussian Process models, because our data has samples from multiple solutions, we address this information to the kernel by adding two RBF kernels, or multiplying 2 RBF kernels and we observe that the model is trying to fit itself onto both the possible solutions. Whereas, we follow the method in Harkonen et al. (2023) to write a physics-informed kernel and gave the additional information to use $n = 1, 2$ states. We again experiment with sum and product of these EPGP kernels. As observed in all the 4 cases in Figure 11, Gaussian Process kernels fail to learn to sample from multiple solutions.

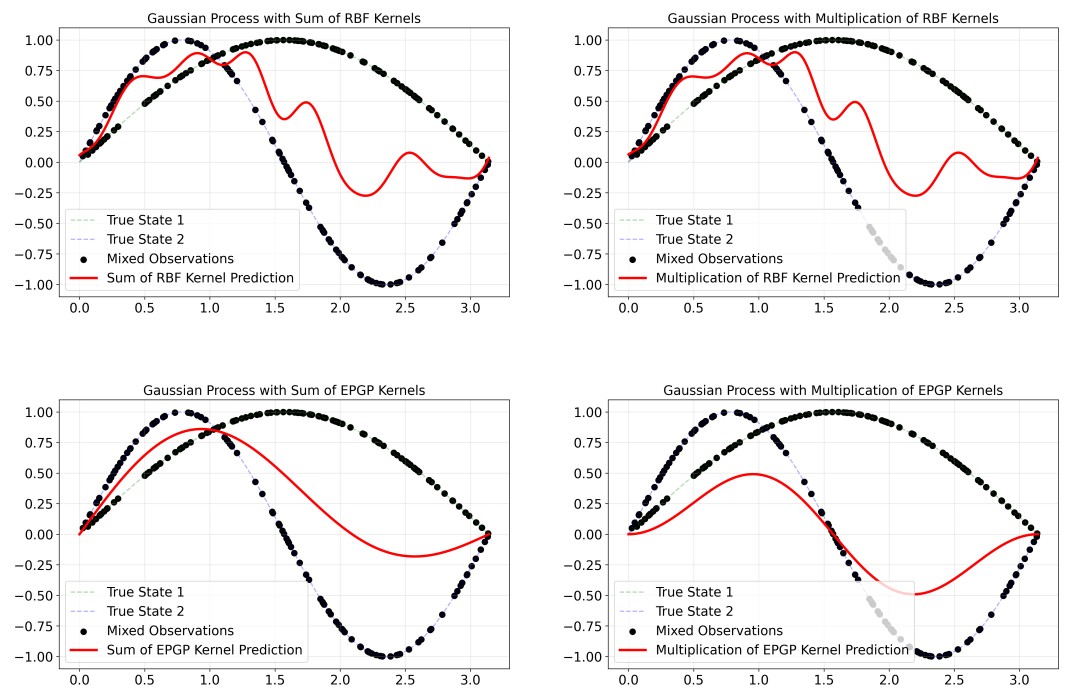

Figure 11: Gaussian Process Experiments for 1D Infinite Potential Well for $n = 1, 2$ eigenstates

## A.7 LIMITATIONS AND FUTURE WORK

As briefly discussed in the main paper, we have demonstrated that PDEDIFF provides a more flexible approach towards incorporating physics in diffusion models as the data is no longer bound to a mesh and the calculation of physics residual can be performed using autograd instead of writing custom finite difference kernels. We have also presented evidence that adding a physics informed loss to a conditional diffusion model can generate solution of PDEs that are more faithful to the underlying physics as compared to using a PINN or a vanilla diffusion model.

Our method lays the foundation for many possible extensions, such as developing a training algorithm to train the diffusion model to learn unseen eigenstates when a particular subset of eigenstates have been provided for training. Another possible future extension could be to develop a data-free approach where no dataset will be required and the denoising step can be guided to generate solutions using the physics informed regularizer.

In a preliminary experiment, we compare the time taken to solve first 4 eigenvalues and eigenfunctions for the 2D infinite potential well Schrodinger equation using an eigensolver (`SciPy`) on a $1000 \times 1000$ grid with the time taken to sample $10^6$ samples from the same grid using PDEDIFF. The eigensolver took almost 5708 seconds whereas PDEDIFF took only around 3126 seconds which is a 1.8 times faster. A potential research direction would be to speed-up the training and sampling time as this could help make the existing PDEDIFF sampler more efficient to generate data on new spatial-temporal coordinates.

Previously, we had also mentioned that scaling this framework to noisy and high-dimensional experimental datasets could potentially accelerate the development of real-time digital simulations of various problems. We believe that our method could be extended to other fields such as to drug discoveries, healthcare and finance, where systems can be modeled as PDEs and these PDEs will have multiple solutions.

## A.8 ABLATION STUDY

To study which weight $c_{res}$ would be best guided our physics-informed PDEDIFF, for each setting, we ran 5 experiments for each value $c_{res} \in \{0, 0.001, 0.005, 0.01, 0.05, 0.1, 1, 10\}$ and average the P-MSE and WD metrics. The best average metrics for non-zero $c_{res}$ for each setting of the Schrodinger equations, and non-homogeneous Helmholtz equation are presented in Table 1 and Table 2 of the main paper.

### A.8.1 ON WEIGHTS FOR RESIDUAL LOSS AND METRICS

In the circle experiments, where the ground truth distribution $\mathbb{P}(Y|X)$ has finitely many modes, PDEDIFF successfully captures all branches of the distribution. PINN, however, treats the problem as a regression task, learning the conditional mean $\mathbb{E}(Y|X)$ and thus fails to accurately predict the $y$-values for given $x$-coordinates. More visualizations of our toy examples that demonstrate this behavior can be found in 9, 8, 7.

As we increase the $c_{res}$ value for the physics constraint, this reduces our training loss as well as the PMSE metric (results shown in Table 5, 6, 7, 8, 9 and better illustrated in Figure 14, 15). Small $c_{res}$ values (0.001, 0.005, 0.01) give low P-MSE and WD metric, with $c_{res} = 0.001$ achieving the best P-MSE and WD for learning 3 wavefunctions for 1D TISE, and $c_{res} = 0.005$ achieving the best metrics for learning 2 wavefunctions for 1D TISE and 1D GP, as shown in Table 5, 7, and 6, respectively. An example of the learned denoising process with $c_{res} = 0.001$ is shown in figure 5.

For 1D Schrodinger cases, a higher $c_{res}$, i.e. $c_{res} = 0.1, 1, 10$ resulted in a flatter curve distribution (figure 13) compared to the distribution learned with a smaller $c_{res} = 0.001, 0.005, 0.01$ (figure 12). One reason might be that strong physics residuals make the distribution converge to the trivial solution where the wavefunction $\psi(x) = 0$ and the energy $E = 0$. This trivial solution also satisfies the Schrodinger equation of the form $\mathcal{H}\psi = E\psi$. In addition, more points tend to concentrate around the first predicted eigenvalue, or also known as mode collapsed. This occurs when we enforce a strong penalty on the physics-informed loss.

Across all settings, the residual weight ($c_{res}$) is balancing the two scenarios: a small value allows for more flexible generation, while for larger values it forces the model to learn a trivial solution. As

| Setting | Unit circle | | Disjoint circles | | Concentric circles | |
|---|---|---|---|---|---|---|
| $c_{res}$ | P | Q | P | Q | P | Q |
| 0 — MSE | 0.389±0.058 | 0.0028±0.0014 | 0.489±0.053 | 0.013±0.0023 | 0.277±0.085 | 0.052±0.014 |
| 0 — WD | 0.475±0.010 | 0.0116±0.0192 | 0.533±0.015 | 0.421±0.055 | 0.907±0.032 | 0.560±0.038 |
| 0.01 — MSE | 0.379±0.085 | 0.0014±0.0006 | 0.486±0.052 | 0.012±0.0038 | 0.274±0.092 | 0.052±0.012 |
| 0.01 — WD | 0.471±0.013 | 0.0852±0.0191 | 0.532±0.015 | 0.417±0.089 | 0.906±0.037 | 0.549±0.030 |

Table 4: Performance comparison between PINN (P) and PDEDIFF (Q) across different problem settings. Here, MSE is as defined in section A.4.1 and WD is the Wasserstein metric for uniform distribution on circles defined in section A.4.2. The error is range of one standard deviation.

observed in Table 5, 6, and 7, increasing $c_{res}$, causes the model to collapse to a single mode, where the PMSE remains small for PDEDIFF, but the Wasserstein distance increases.

Similarly, for the 2D Schrodinger cases, we can see a similar trend with respect to $c_{res}$ in Table 8, 9. For the harder 4 wave case, PDEDIFF achieves a better Wasserstein metric as compared to the other methods. Since the "average" eigenstate for the 4 waves is the degenerate state, i.e., the eigenstates with the same energy $((1, 2), (2, 1))$, PINN is able to give a much lower P-MSE value, but we can see from the Wasserstein metric, the samples generated have collapsed onto this average state and is not able to generate a mixture of all the learned states. PDEDIFF in this case is still able to prevent this catastrophic mode collapse and generate a good mixture of eigenstates.

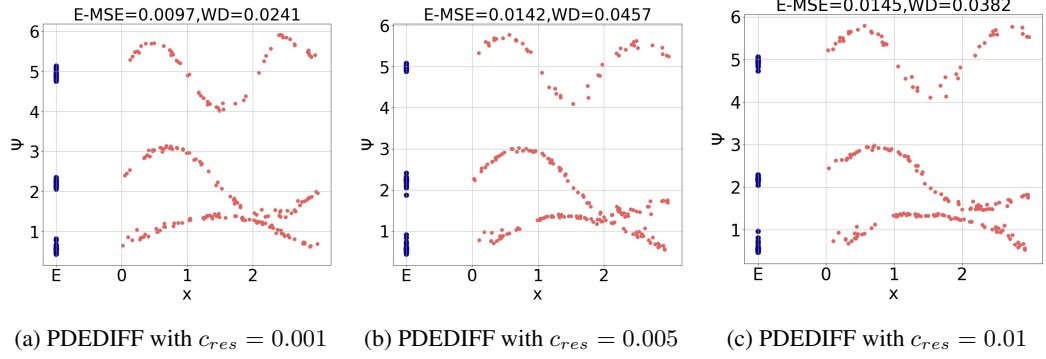

(a) PDEDIFF with $c_{res} = 0.001$     (b) PDEDIFF with $c_{res} = 0.005$     (c) PDEDIFF with $c_{res} = 0.01$

Figure 12: Generated samples by physics-informed PDEDIFF with different $c_{res}$ values for 1D Schrodinger equation in infinite potential well.

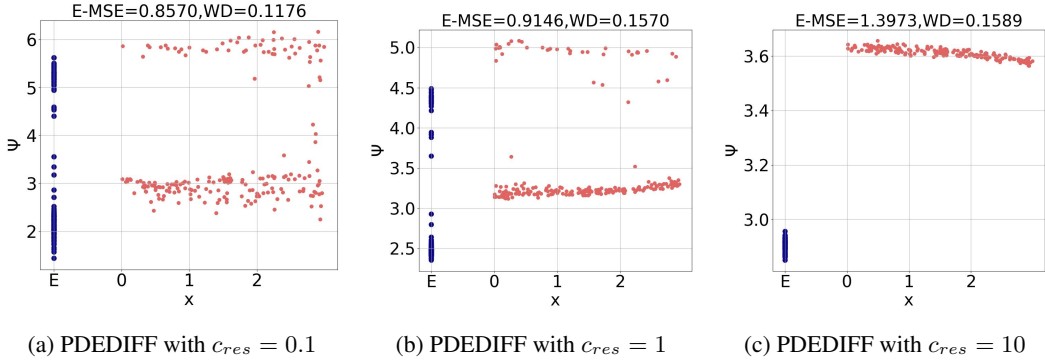

(a) PDEDIFF with $c_{res} = 0.1$     (b) PDEDIFF with $c_{res} = 1$     (c) PDEDIFF with $c_{res} = 10$

Figure 13: Generated samples by physics-informed PDEDIFF with different $c_{res}$ values for 1D Schrodinger equation in infinite potential well.

| Setting | | 1D (2 waves: $n = 1, 2$) | | | |
|---|---|---|---|---|---|
| $c_{res}$ | | P | D | N | Q |
| 0 | PMSE | $0.822 \pm 0.016$ | $0.9323 \pm 0.0225$ | $0.8090 \pm 0.0236$ | $0.0168 \pm 0.010$ |
| | WD | $0.539 \pm 0.022$ | $0.1572 \pm 0.00601$ | $0.5440 \pm 0.0147$ | $0.0863 \pm 0.059$ |
| 0.001 | PMSE | $0.974 \pm 0.247$ | $0.9351 \pm 0.0251$ | $0.9600 \pm 0.1299$ | $0.0137 \pm 0.013$ |
| | WD | $0.423 \pm 0.052$ | $0.1572 \pm 0.00602$ | $0.4460 \pm 0.0442$ | $0.0865 \pm 0.063$ |
| 0.005 | PMSE | $1.052 \pm 0.329$ | $0.9350 \pm 0.0249$ | $1.0300 \pm 0.1924$ | $0.0146 \pm 0.011$ |
| | WD | $0.244 \pm 0.031$ | $0.1570 \pm 0.00602$ | $0.3500 \pm 0.0421$ | $0.0862 \pm 0.061$ |
| 0.01 | PMSE | $1.106 \pm 0.371$ | $0.9348 \pm 0.0247$ | $1.1082 \pm 0.2024$ | $0.0392 \pm 0.032$ |
| | WD | $0.313 \pm 0.020$ | $0.1572 \pm 0.00602$ | $0.3077 \pm 0.0349$ | $0.113 \pm 0.037$ |
| 0.05 | PMSE | $1.923 \pm 0.456$ | $0.9370 \pm 0.0264$ | $1.3800 \pm 0.2067$ | $0.239 \pm 0.157$ |
| | WD | $0.327 \pm 0.069$ | $0.1570 \pm 0.00609$ | $0.2300 \pm 0.0188$ | $0.152 \pm 0.057$ |
| 0.1 | PMSE | $3.414 \pm 1.49$ | $0.9400 \pm 0.0274$ | $1.5200 \pm 0.2919$ | $0.748 \pm 0.411$ |
| | WD | $0.587 \pm 0.130$ | $0.1580 \pm 0.00593$ | $0.2270 \pm 0.0350$ | $0.153 \pm 0.057$ |
| 1 | PMSE | $3325 \pm 2996$ | $0.9390 \pm 0.0412$ | $1.7900 \pm 0.4569$ | $1.05 \pm 0.317$ |
| | WD | $0.306 \pm 0.206$ | $0.1730 \pm 0.0100$ | $0.1260 \pm 0.0046$ | $0.188 \pm 0.476$ |
| 10 | PMSE | $3540 \pm 2741$ | $0.9910 \pm 0.0376$ | $2.6000 \pm 2.1009$ | $2.05 \pm 1.753$ |
| | WD | $0.236 \pm 0.110$ | $0.1670 \pm 0.0193$ | $0.1280 \pm 0.0106$ | $0.170 \pm 0.009$ |

Table 5: Performance comparison between PINN (P), DeepONet (D), Physic-informed Neural Operator (N), and PDEDIFF (Q) for 1D (2 waves).

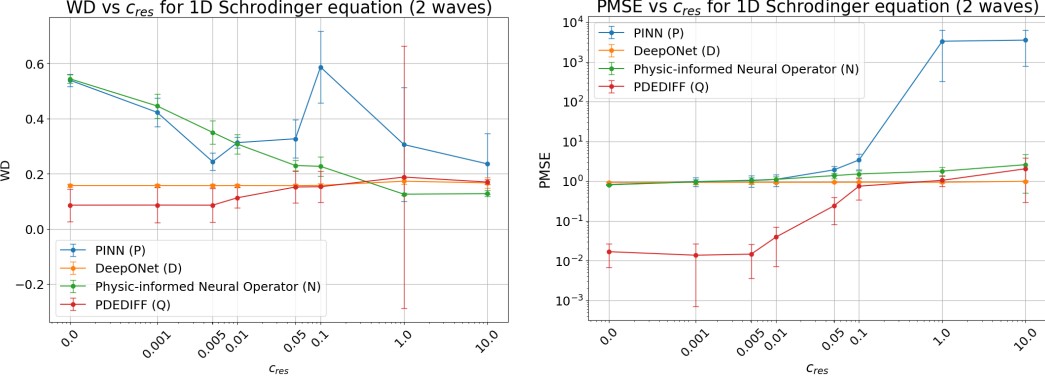

Figure 14: Ablation study results from Table 5 with error bar for 1D Schrodinger equation (2 waves). PDEDIFF achieves lowest PMSE and WD metric compare with PINN, Physics-informed Neural Operator and DeepONet for $c_{res} \leq 0.1$ values. Overall, PDEDIFF with $c_{res} = 0.005$ gives the lowest PMSE and WD metrics.

| Setting | | 1D (3 waves: $n = 1, 2, 3$) | | | |
|---|---|---|---|---|---|
| $c_{res}$ | | P | D | N | Q |
| 0 | PMSE | $0.371 \pm 0.031$ | $4.37 \pm 0.029$ | $0.3330 \pm 0.0337$ | $0.0218 \pm 0.013$ |
| | WD | $0.737 \pm 0.023$ | $0.114 \pm 0.00275$ | $0.7760 \pm 0.0331$ | $0.0549 \pm 0.009$ |
| 0.001 | PMSE | $0.460 \pm 0.435$ | $4.37 \pm 0.029$ | $0.4450 \pm 0.0349$ | $0.0123 \pm 0.012$ |
| | WD | $0.422 \pm 0.151$ | $0.114 \pm 0.00275$ | $0.4700 \pm 0.1350$ | $0.0499 \pm 0.019$ |
| 0.005 | PMSE | $0.521 \pm 0.079$ | $4.37 \pm 0.029$ | $0.5250 \pm 0.0374$ | $0.0299 \pm 0.011$ |
| | WD | $0.315 \pm 0.088$ | $0.114 \pm 0.00275$ | $0.3370 \pm 0.0827$ | $0.0716 \pm 0.014$ |
| 0.01 | PMSE | $0.534 \pm 0.167$ | $4.37 \pm 0.029$ | $0.5831 \pm 0.0449$ | $0.0347 \pm 0.006$ |
| | WD | $0.266 \pm 0.064$ | $0.114 \pm 0.00275$ | $0.2907 \pm 0.0643$ | $0.0838 \pm 0.022$ |
| 0.05 | PMSE | $1.330 \pm 0.528$ | $4.38 \pm 0.0307$ | $0.8140 \pm 0.0964$ | $0.532 \pm 0.222$ |
| | WD | $0.399 \pm 0.154$ | $0.114 \pm 0.00287$ | $0.1920 \pm 0.0316$ | $0.176 \pm 0.130$ |
| 0.1 | PMSE | $7.63 \pm 6.148$ | $4.38 \pm 0.0335$ | $0.9540 \pm 0.1395$ | $0.807 \pm 0.108$ |
| | WD | $0.503 \pm 0.132$ | $0.115 \pm 0.00327$ | $0.1270 \pm 0.0045$ | $0.111 \pm 0.013$ |
| 1 | PMSE | $6989 \pm 6330$ | $4.41 \pm 0.0335$ | $1.0800 \pm 0.1151$ | $0.971 \pm 0.171$ |
| | WD | $0.224 \pm 0.149$ | $0.113 \pm 0.00676$ | $0.1320 \pm 0.0251$ | $0.145 \pm 0.059$ |
| 10 | PMSE | $7552 \pm 5669$ | $4.20 \pm 0.138$ | $2.1200 \pm 2.5231$ | $1.56 \pm 1.61$ |
| | WD | $0.154 \pm 0.056$ | $0.165 \pm 0.0621$ | $0.1260 \pm 0.0161$ | $0.143 \pm 0.0241$ |

Table 6: Performance comparison between PINN (P), DeepONet (D), Physic-informed Neural Operator (N), and PDEDIFF (Q) for 1D (3 waves).

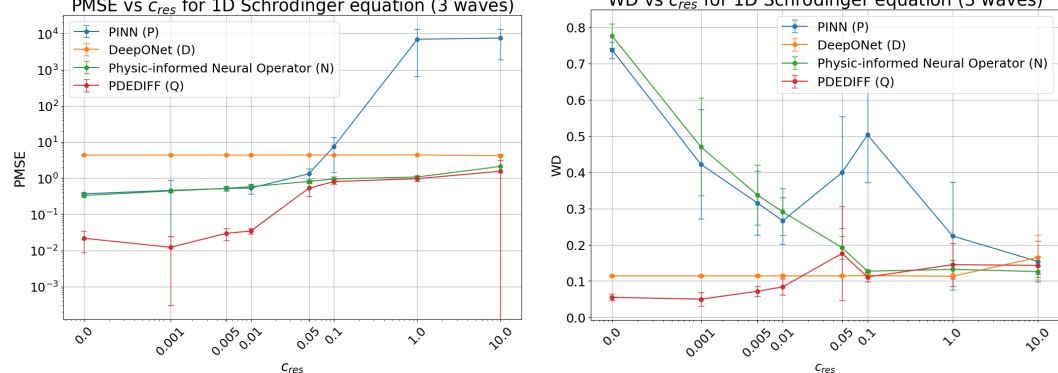

Figure 15: Ablation study results from Table 6 with error bar for 1D Schrodinger equation (3 waves). PDEDIFF achieves lowest PMSE and WD metric compare with PINN, Physics-informed Neural Operator and DeepONet for small $c_{res}$ values. Overall, PDEDIFF with $c_{res} = 0.001$ gives the lowest PMSE and WD metrics.

| Setting | | 1D GP (3 waves: $n = 1, 2, 3$) | | | |
|---|---|---|---|---|---|
| $c_{res}$ | | P | D | N | Q |
| 0 | PMSE | $1.550 \pm 0.804$ | $0.868 \pm 0.004$ | $0.0685 \pm 0.00263$ | $0.140 \pm 0.007$ |
| | WD | $2.036 \pm 1.222$ | $1.431 \pm 0.027$ | $1.180 \pm 0.0449$ | $0.195 \pm 0.043$ |
| 0.001 | PMSE | $1.291 \pm 0.479$ | $0.867 \pm 0.00487$ | $0.0691 \pm 0.00260$ | $0.126 \pm 0.019$ |
| | WD | $2.418 \pm 1.084$ | $1.431 \pm 0.0274$ | $1.185 \pm 0.0451$ | $0.194 \pm 0.033$ |
| 0.005 | PMSE | $1.550 \pm 0.804$ | $0.868 \pm 0.00451$ | $0.0708 \pm 0.00243$ | $0.118 \pm 0.028$ |
| | WD | $2.040 \pm 1.222$ | $1.430 \pm 0.0274$ | $1.203 \pm 0.0472$ | $0.338 \pm 0.186$ |
| 0.01 | PMSE | $1.550 \pm 0.804$ | $0.868 \pm 0.00438$ | $0.0732 \pm 0.00222$ | $0.139 \pm 0.032$ |
| | WD | $2.036 \pm 1.222$ | $1.431 \pm 0.0274$ | $1.223 \pm 0.0533$ | $0.333 \pm 0.187$ |
| 0.05 | PMSE | $1.550 \pm 0.804$ | $0.869 \pm 0.00440$ | $0.0874 \pm 0.00399$ | $0.134 \pm 0.018$ |
| | WD | $2.040 \pm 1.222$ | $1.430 \pm 0.0276$ | $1.236 \pm 0.0903$ | $0.937 \pm 0.734$ |
| 0.1 | PMSE | $1.550 \pm 0.804$ | $0.871 \pm 0.00507$ | $0.0930 \pm 0.00923$ | $0.189 \pm 0.089$ |
| | WD | $2.040 \pm 1.222$ | $1.430 \pm 0.0276$ | $1.144 \pm 0.175$ | $0.644 \pm 0.453$ |
| 1 | PMSE | $1.550 \pm 0.804$ | $0.898 \pm 0.00864$ | $0.1016 \pm 0.00198$ | $0.248 \pm 0.137$ |
| | WD | $2.040 \pm 1.222$ | $1.430 \pm 0.0260$ | $1.899 \pm 0.889$ | $1.100 \pm 0.565$ |
| 10 | PMSE | $1.550 \pm 0.804$ | $0.915 \pm 0.0427$ | $0.1016 \pm 0.00197$ | $0.286 \pm 0.204$ |
| | WD | $2.040 \pm 1.222$ | $1.120 \pm 0.0841$ | $2.992 \pm 0.414$ | $0.453 \pm 0.072$ |

Table 7: Performance comparison between PINN (P), DeepONet (D), Physic-informed Neural Operator (N), and PDEDIFF (Q) for the 1D GP (3 waves) setting. Lower values indicate better agreement with the ground truth wavefunctions.

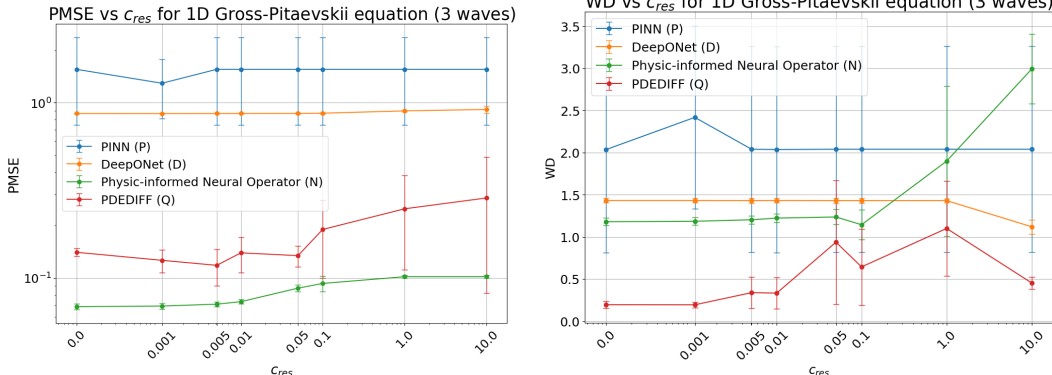

Figure 16: Ablation study results from Table 7 with error bar for 1D Gross-Pitaevskii equation (3 waves). PDEDIFF achieves lowest WD metric compare with PINN, Physics-informed Neural Operator and DeepONet for all $c_{res} \leq 10$ values. Physics-informed neural operator converges to a single mode to achieve the small PMSE metric but failed to learn the whole multi-modal distribution with all 3 wavefunctions. This showed in Figure 4 and the high WD metric.

| Setting | | 2D (3 waves: $(1, 1), (2, 1), (2, 2)$) | | | |
|---------|------|--------------------|-------------------|-------------------|-------------------|
| $c_{res}$ | | P | D | N | Q |
| 0 | PMSE | $1.253 \pm 0.147$ | $7.3269 \pm 0.2681$ | $1.2543 \pm 0.1392$ | $1.222 \pm 0.071$ |
|   | WD | $0.378 \pm 0.072$ | $0.4419 \pm 0.0584$ | $0.2967 \pm 0.0301$ | $0.169 \pm 0.037$ |
| 0.001 | PMSE | $1.239 \pm 0.098$ | $7.3264 \pm 0.2682$ | $1.3980 \pm 0.6033$ | $1.194 \pm 0.097$ |
|   | WD | $0.340 \pm 0.075$ | $0.4420 \pm 0.0584$ | $0.5020 \pm 0.2341$ | $0.155 \pm 0.037$ |
| 0.005 | PMSE | $1.229 \pm 0.061$ | $7.3300 \pm 0.2676$ | $2.0700 \pm 0.6581$ | $1.201 \pm 0.098$ |
|   | WD | $0.363 \pm 0.100$ | $1.1800 \pm 1.6280$ | $0.8260 \pm 0.1594$ | $0.187 \pm 0.042$ |
| 0.01 | PMSE | $1.184 \pm 0.062$ | $7.3259 \pm 0.2675$ | $4.7610 \pm 7.9025$ | $1.174 \pm 0.098$ |
|   | WD | $0.368 \pm 0.076$ | $0.4419 \pm 0.0584$ | $0.8188 \pm 0.2973$ | $0.173 \pm 0.047$ |
| 0.05 | PMSE | $1.004 \pm 0.087$ | $7.3200 \pm 0.2651$ | $245373 \pm 548545$ | $0.923 \pm 0.095$ |
|   | WD | $0.444 \pm 0.058$ | $0.4420 \pm 0.0585$ | $1.09 \pm 0.4884$ | $0.238 \pm 0.047$ |
| 0.1 | PMSE | $0.793 \pm 0.049$ | $7.3300 \pm 0.2725$ | $21437 \pm 3047.6$ | $0.769 \pm 0.145$ |
|   | WD | $0.502 \pm 0.083$ | $0.4420 \pm 0.0587$ | $1.31 \pm 0.4834$ | $0.264 \pm 0.0599$ |
| 1 | PMSE | $49.531 \pm 3.639$ | $7.3700 \pm 0.2144$ | $571194 \pm 883341$ | $1.103 \pm 1.247$ |
|   | WD | $0.239 \pm 0.047$ | $0.4390 \pm 0.0536$ | $1.33 \pm 0.3529$ | $0.338 \pm 0.083$ |
| 10 | PMSE | $1474.578 \pm 293.978$ | $7.2100 \pm 0.1260$ | $16423 \pm 2072.9$ | $3.059 \pm 2.304$ |
|   | WD | $0.169 \pm 0.044$ | $0.4540 \pm 0.0143$ | $0.690 \pm 0.1398$ | $0.248 \pm 0.048$ |

Table 8: Results for the 2D (3 waves: $(1, 1), (2, 1), (2, 2)$) experiment, comparing PINN (P), Deep-ONet (D), Neural Operator (N), and PDEDIFF (Q).

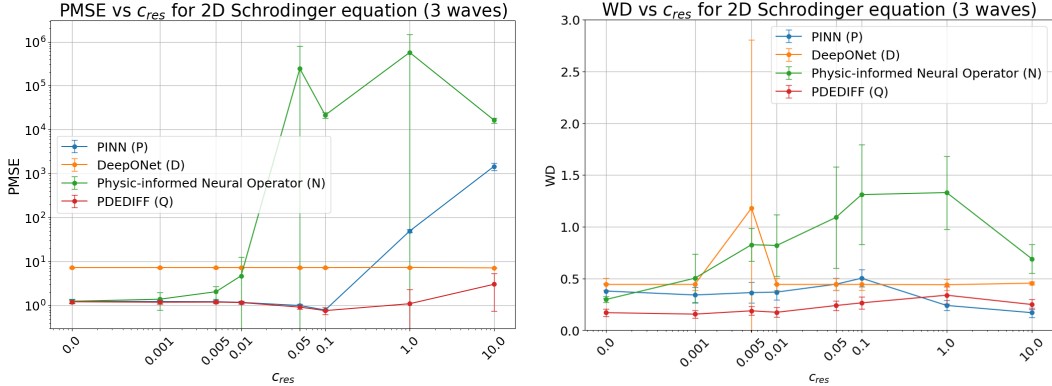

Figure 17: Ablation study results from Table 8 with error bar for 2D Schrodinger equation (3 waves). PDEDIFF achieves lowest WD and PMSE metric compare with PINN, Physics-informed Neural Operator and DeepONet for all $c_{res} \leq 0.1$ values. PDEDIFF with $c_{res} = 0.1$ gives the lowest PMSE metric and PDEDIFF with $c_{res} = 0.001$ gives the lowest WD metric.

| Setting | | 2D (4 waves: $(1,1), (1,2), (2,1), (2,2)$) | | | |
|---|---|---|---|---|---|
| $c_{res}$ | | P | D | N | Q |
| 0 | PMSE | $1.250 \pm 0.059$ | $7.661 \pm 0.539$ | $1.332 \pm 0.197$ | $1.309 \pm 0.073$ |
| | WD | $0.482 \pm 0.073$ | $0.4846 \pm 0.0709$ | $0.300 \pm 0.103$ | $0.150 \pm 0.042$ |
| 0.001 | PMSE | $1.243 \pm 0.055$ | $7.659 \pm 0.539$ | $0.933 \pm 0.305$ | $1.338 \pm 0.074$ |
| | WD | $0.477 \pm 0.078$ | $0.4846 \pm 0.0708$ | $0.513 \pm 0.104$ | $0.124 \pm 0.051$ |
| 0.005 | PMSE | $1.223 \pm 0.038$ | $7.660 \pm 0.539$ | $135492 \pm 302955$ | $1.390 \pm 0.147$ |
| | WD | $0.481 \pm 0.077$ | $0.4850 \pm 0.0708$ | $1.07 \pm 0.647$ | $0.202 \pm 0.118$ |
| 0.01 | PMSE | $1.181 \pm 0.041$ | $7.661 \pm 0.539$ | $129839 \pm 289801$ | $1.414 \pm 0.192$ |
| | WD | $0.527 \pm 0.105$ | $0.4846 \pm 0.0708$ | $1.010 \pm 0.629$ | $0.194 \pm 0.103$ |
| 0.05 | PMSE | $1.006 \pm 0.042$ | $7.660 \pm 0.539$ | $1209.46 \pm 2686.2$ | $1.088 \pm 0.211$ |
| | WD | $0.592 \pm 0.094$ | $0.4850 \pm 0.0708$ | $1.22 \pm 0.377$ | $0.386 \pm 0.114$ |
| 0.1 | PMSE | $0.790 \pm 0.041$ | $7.660 \pm 0.539$ | $3388.96 \pm 7461.78$ | $0.928 \pm 0.109$ |
| | WD | $0.597 \pm 0.069$ | $0.4850 \pm 0.0708$ | $1.05 \pm 0.306$ | $0.326 \pm 0.098$ |
| 1 | PMSE | $21.005 \pm 1.826$ | $7.670 \pm 0.400$ | $46626.9 \pm 44388$ | $0.815 \pm 0.165$ |
| | WD | $0.211 \pm 0.035$ | $0.4840 \pm 0.0675$ | $1.41 \pm 0.354$ | $0.301 \pm 0.032$ |
| 10 | PMSE | $460.817 \pm 56.397$ | $7.290 \pm 0.146$ | $1317743 \pm 2756065$ | $0.808 \pm 0.201$ |
| | WD | $0.160 \pm 0.046$ | $0.4580 \pm 0.0135$ | $1.44 \pm 0.360$ | $0.392 \pm 0.162$ |

Table 9: Results for 2D (4 waves: $(1,1), (1,2), (2,1), (2,2)$) experiment, comparing PINN (P), DeepONet (D), Neural Operator (N), and PDEDIFF (Q).

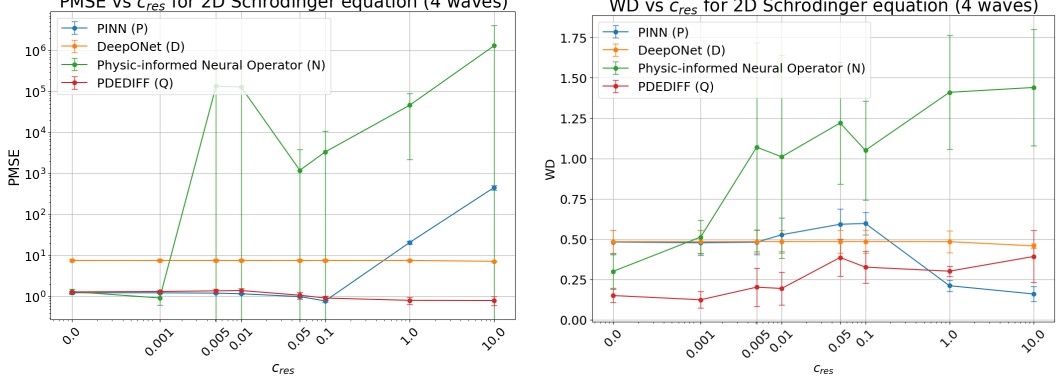

Figure 18: Ablation study results from Table 9 with error bar for 2D Schrodinger equation (4 waves). PDEDIFF achieves lowest WD metric compare with PINN, Physics-informed Neural Operator and DeepONet for all $c_{res} \leq 0.1$ values. PINN with $c_{res} = 10$ gives the lowest PMSE metric and PDEDIFF with $c_{res} = 0.001$ gives the lowest WD metric.

| Setting | | 2D (4 waves: $(1,1), (2,1), (3,1), (4,1)$) | | | |
|---------|------|-----------------------|-----------------------|-----------------------|------------------------|
| $c_{res}$ | | P | D | N | Q |
| 0 | PMSE | $1.1333 \pm 0.0696$ | $22.342 \pm 2.3903$ | $1.1984 \pm 0.1170$ | $1.1004 \pm 0.0490$ |
| | WD | $0.5973 \pm 0.1604$ | $0.5127 \pm 0.1082$ | $0.3442 \pm 0.1107$ | $0.22174 \pm 0.1484$ |
| 0.001 | PMSE | $1.1299 \pm 0.05626$ | $22.342 \pm 2.3903$ | $1.2570 \pm 0.5490$ | $1.0905 \pm 0.0559$ |
| | WD | $0.6233 \pm 0.0560$ | $0.5127 \pm 0.1082$ | $0.58746 \pm 0.1358$ | $0.20494 \pm 0.0891$ |
| 0.005 | PMSE | $1.9343 \pm 0.7272$ | $22.342 \pm 2.39033$ | $2.2754 \pm 2.1479$ | $1.0326 \pm 0.1810$ |
| | WD | $0.4815 \pm 0.0074$ | $0.5128 \pm 0.1083$ | $0.67119 \pm 0.2896$ | $0.38876 \pm 0.2228$ |
| 0.01 | PMSE | $1.6699 \pm 0.8890$ | $22.342 \pm 2.3903$ | $8328.97 \pm 18620$ | $0.9246 \pm 0.1270$ |
| | WD | $0.4702 \pm 0.0269$ | $0.5127 \pm 0.1082$ | $0.8739 \pm 0.5174$ | $0.5585 \pm 0.1533$ |
| 0.05 | PMSE | $2.0404 \pm 1.3113$ | $22.344 \pm 2.3907$ | $1774100 \pm 3922300$ | $0.7339 \pm 0.1237$ |
| | WD | $0.46002 \pm 0.0199$ | $0.51276 \pm 0.1082$ | $1.1972 \pm 0.4778$ | $0.4753 \pm 0.0588$ |
| 0.1 | PMSE | $2.2239 \pm 1.5007$ | $22.350 \pm 2.3901$ | $604850 \pm 1335200$ | $0.6149 \pm 0.1649$ |
| | WD | $0.4576 \pm 0.0203$ | $0.5127 \pm 0.1081$ | $1.3754 \pm 0.5193$ | $0.4431 \pm 0.0244$ |
| 1 | PMSE | $15.223 \pm 16.508$ | $22.362 \pm 2.2990$ | $31462.9 \pm 69532.2$ | $1.50 \pm 2.2162$ |
| | WD | $0.5161 \pm 0.0757$ | $0.5129 \pm 0.1050$ | $1.0066 \pm 0.3278$ | $0.4436 \pm 0.0018$ |
| 10 | PMSE | $637.61 \pm 197.99$ | $20.565 \pm 0.5769$ | $863.57 \pm 375.39$ | $4.3467 \pm 8.2002$ |
| | WD | $0.6188 \pm 0.1821$ | $0.4660 \pm 0.0136$ | $0.7310 \pm 0.0744$ | $0.4640 \pm 0.0443$ |

Table 10: Results for the 2D (4 waves: $(1,1), (2,1), (3,1), (4,1)$) experiment, comparing PINN (P), DeepONet (D), Neural Operator (N), and PDEDIFF (Q).

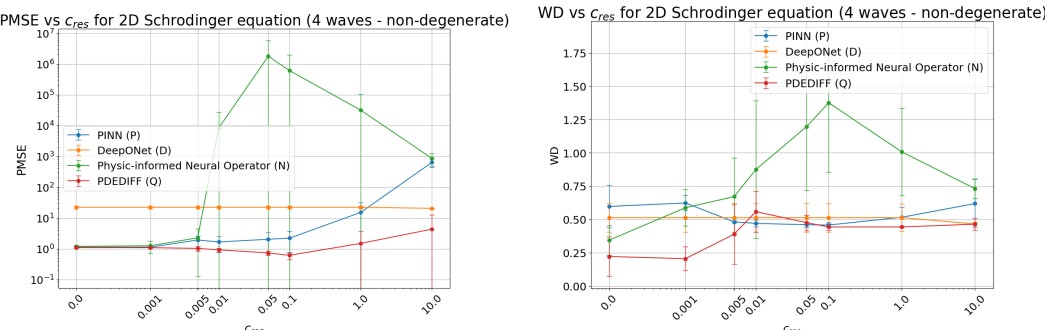

Figure 19: Ablation study results from Table 10 with error bar for 2D Schrodinger equation (4 waves). PDEDIFF achieves lowest PMSE metric compare with PINN, Physics-informed Neural Operator and DeepONet for all $c_{res}$ values. PDEDIFF with $c_{res} = 0.001$ gives the lowest WD metric.

| Setting | | Helmholtz (2 parameters: $k \in \{\pi, 4\pi\}$) | | | |
|---|---|---|---|---|---|
| $c_{res}$ | | P | D | N | Q |
| 0 | PMSE | $0.800 \pm 0.110$ | $21.35 \pm 2.67$ | $0.753 \pm 0.217$ | $0.254 \pm 0.153$ |
| | WD | $0.107 \pm 0.0401$ | $0.168 \pm 0.00103$ | $0.111 \pm 0.0425$ | $0.0915 \pm 0.0052$ |
| 0.001 | PMSE | $0.746 \pm 0.315$ | $21.35 \pm 2.67$ | $1.00 \pm 1.62$ | $0.227 \pm 0.0867$ |
| | WD | $0.108 \pm 0.0297$ | $0.168 \pm 0.00103$ | $0.106 \pm 0.0463$ | $0.0913 \pm 0.00615$ |
| 0.005 | PMSE | $0.918 \pm 0.164$ | $21.4 \pm 2.67$ | $99.5 \pm 178.81$ | $0.320 \pm 0.133$ |
| | WD | $0.102 \pm 0.00921$ | $0.168 \pm 0.00103$ | $0.303 \pm 0.153$ | $0.0874 \pm 0.00455$ |
| 0.01 | PMSE | $1.18 \pm 0.476$ | $21.35 \pm 2.67$ | $0.909 \pm 1.01$ | $0.570 \pm 0.187$ |
| | WD | $0.109 \pm 0.0234$ | $0.168 \pm 0.00103$ | $0.160 \pm 0.148$ | $0.0886 \pm 0.00580$ |
| 0.05 | PMSE | $38.4 \pm 17.41$ | $21.6 \pm 2.76$ | $140 \pm 126.08$ | $2.740 \pm 2.170$ |
| | WD | $0.0866 \pm 0.00753$ | $0.168 \pm 0.001034$ | $0.0925 \pm 0.0121$ | $0.0819 \pm 0.00715$ |
| 0.1 | PMSE | $48.7 \pm 19.58$ | $23.4 \pm 3.02$ | $159 \pm 145.52$ | $4.430 \pm 0.566$ |
| | WD | $0.0875 \pm 0.00734$ | $0.167 \pm 0.001047$ | $0.0937 \pm 0.0126$ | $0.0803 \pm 0.00419$ |
| 1 | PMSE | $74.6 \pm 27.11$ | $11.1 \pm 9.56$ | $196 \pm 158.64$ | $5.510 \pm 0.581$ |
| | WD | $0.0888 \pm 0.00670$ | $0.173 \pm 0.03096$ | $0.0987 \pm 0.0134$ | $0.0827 \pm 0.00703$ |
| 10 | PMSE | $89.9 \pm 25.42$ | $0.557 \pm 0.000874$ | $458 \pm 246.35$ | $6.540 \pm 1.234$ |
| | WD | $0.0901 \pm 0.00637$ | $0.158 \pm 0.0280$ | $0.169 \pm 0.0566$ | $0.0746 \pm 0.00446$ |

Table 11: Results for the Helmholtz (2 parameters) experiment, comparing PINN (P), DeepONet (D), Physics-informed Neural Operator (N), and PDEDIFF (Q).

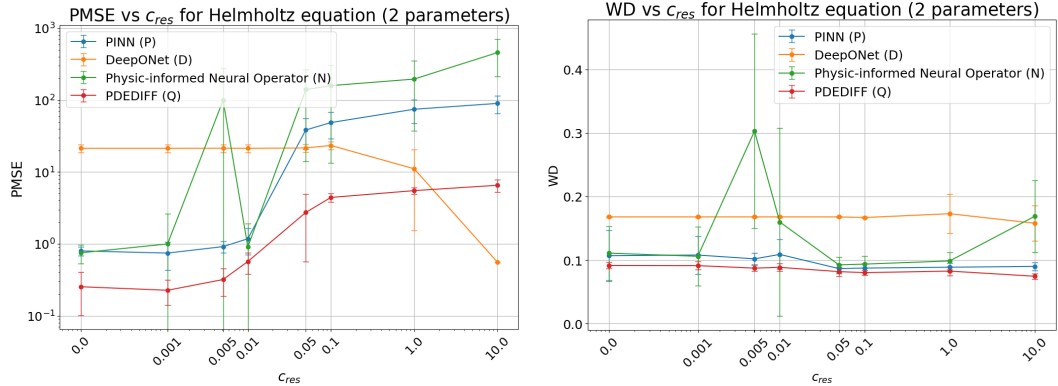

Figure 20: Ablation study results from Table 11 with error bar for Helmholtz equation (4 waves). PDEDIFF achieves lowest PMSE metric compare with PINN, Physics-informed Neural Operator and DeepONet for all $c_{res} \leq 1$ values. PDEDIFF with $c_{res} = 0.001$ gives the lowest PMSE metric and PDEDIFF with $c_{res} = 10$ gives the lowest WD metric.

