# OpenReview forum: "Physics-Informed Conditional Diffusion for Multi-Modal PDEs"
_ICLR.cc/2026/Conference — Submitted to ICLR 2026_

### Official Review · Reviewer_A7D5 · 2025-10-16

**Soundness:** 3
**Presentation:** 3
**Contribution:** 2
**Rating:** 4
**Confidence:** 2

**Summary:**

The paper suggests an approach for physics (i.e. PDE) informed diffusion models that work on PDE systems that are not well-posed. This approach has several advantages: independences of input positions, i.e. being meshless;

**Strengths:**

The approach is sound and the paper is mostly understandable. There is novelty (despite being a combination of well-known techniques) in the paper that would in general make it a fit for ICLR. Enough additional information is in the appendixes.

**Weaknesses:**

My main problems is with the experiments:
 - I do not understand how you can reasonably compare to PINNs, which are a regression technique and thereby, kind of by definition, present you just a mean function.
 - Your experiments are all linear or almost linear equations (no systems, only weak non-linearities). It sounds somewhat restrictive to avoid non-linear equations. Furthermore, for linear PDE systems, there is a strong linear of research for generative models using Gaussian Processes, which might very well be superior to the presented approach in the 3 out of 4 test cases.
(This is despite the nice choise of evaluation metrics.)
I would need a clear indication of me misreading this problem to change my rating.

There are some minor points with presentations. While the paper is understandable, it does is not easy to read the paper and it would benefit from stylistic improvements.

**Questions:**

l. 223 definite article in "the probability distribution"?
in (10), do you need to reconstruct $\hat\Psi_0$ each time in training?
l. 306: Which eigenvalues?
l. 323: Which eigenstates?
l. 351: Which nodes?

---

> ### Author Response · Authors · 2025-11-21
>
> We thank all reviewers for their time and constructive feedback on our submission. We are encouraged by the positive remarks that recognize our motivation to address multi-modal PDE modeling, which has been a long standing limitation of physics-informed systems. We are glad that the reviewers found the experimental results motivating and recognized our method’s ability to generate diverse, physically consistent solutions. We also appreciate that the reviewers found our paper understandable with sufficient explanations of the experimental setups. We carefully address all concerns below and also pose clarifying questions to better understand how we could improve the scope, evaluation, and clarity of our work in future versions.
>
> **Non-linear PDE.** In our paper, we have discussed a non-linear case in the family of Schrodinger Equation where we have experimented on Gross-Pitaevskii Equation in Section 5.2. This equation is non-linear due to the presence on $|\\psi|^2$ term in the differential operator. Furthermore, to extend our study, we have also conducted a preliminary experiment on Burgers' Equation and the results are below for the same setup as the other experiments in the paper.
>
> | Setting | **P**        | **Q**        |
> |---------|----------------------------------|----------------------------------|
> | 0       |$0.0516 \\pm 0.0024$              |$0.0149 \\pm 0.0025$             |
> | 0.001   | $0.0685 \\pm 0.0119$             | $0.0148 \\pm 0.0023$              |
>
> _**Table:** Preliminary performance comparison between PINN (P) and PDEDIFF (Q) for Burgers' Equation on a specific trajectory with two different viscosities. Values are P-MSE (lower is better)._
>
>
> **Grammatical fix.** We will update the grammatical error on line 223 with "a probability distribution"
>
> **In (10), do you need to reconstruct $\\mathbf{\\hat{\\Psi}\_0}$ each time in training?**
>
> We do compute $\\mathbf{\\hat{\Psi}\_0}$  at every epoch in during training according to Algorithm 1 (specifically in step 10), we then compute the gradients of the loss function (in equation (11)) to update $CCED\_{\\theta}$. In the next epoch, we will compute $\\mathbf{\\hat{\\Psi}\_0}$ via the updated $CCED\_{\\theta}$ and repeat until convergence or a certain number of epochs.
>
> **Line 306 clarification.** The eigenvalues mentioned here are the eigenvalues corresponding to the solution function. The updated statement should clear the ambiguity: Figure 1 demonstrates how the denoising model converts random noise into the corresponding solutions functions and their corresponding eigenvalues/parameters.
>
> **Line 323 clarification.** The eigenstates refer to the ground truth solutions from the dataset, such as the first three eigenstates of 1D infinite potential well.
>
> **Line 351 clarification.** The modes refer to the possible eigenvalues/parameters that are a part of the training dataset.
>
> **Question to Reviewer:** As you mentioned that gaussian process are better suited to solve linear PDE systems, we would appreciate it if you could be more detailed or point us to any relevant references for the mentioned methods. We are happy to consider them as alternative baselines.

---

> ### Comment · Reviewer_A7D5 · 2025-11-22
>
> I do not find my main weaknesses adressed.
>
> Regarding the GP comparison: I do not want to make this a condition for acceptance. For completeness sake:
>  - Jidling et al., Linearly constrained Gaussian processes, NeurIPS 2017
>  - Harkonen et al., Gaussian process priors for systems of linear partial differential equations with constant coefficients, ICML 2023
> (And papers cited therein.)

---

> > ### Author Response · Authors · 2025-11-27
> >
> > In the field of Scientific Machine Learning, a common approach is to learn a single possible solution of a PDE system from partially observed data. Yet many physical systems could have multiple solutions with respect to different parameters in the system, such as Helmholtz's sound frequency or Burger's viscosity. Our goal is to address this possible multiplicity in the system. PINN is one of the state-of-the-art baselines for learning-based approaches, which is the reason we compared our method with PINN. We also extend our study to compare against other scientific ML approaches such as Fourier Neural Operator, DeepONet, and their physics-informed versions (see updated results in the table below).
> >
> > | Setting | Model | **Vanilla PMSE** | **Vanilla WD** | **Physics-Informed PMSE** | **Physics-Informed WD** |
> > |--------|--------|------------------|----------------|----------------------------|--------------------------|
> > | **1D (2 states)** | P | 0.822 | 0.539 | 1.052 | 0.344 |
> > |  | D | 0.932 | 0.157 | 0.935 | 0.157 |
> > |  | N | 0.809 | 0.544 | 0.960 | 0.446 |
> > |  | Q | 0.017 | 0.086 | **0.015** | **0.086** |
> > | **1D (3 states)** | P | 0.371 | 0.737 | 0.460 | 0.442 |
> > |  | D | 0.438 | 0.114 | 0.437 | 0.114 |
> > |  | N | 0.333 | 0.776 | 0.445 | 0.470 |
> > |  | Q | 0.022 | 0.055 | **0.012** | **0.050** |
> > | **1D GP (3 states)** | P | 1.550 | 2.036 | 1.291 | 2.418 |
> > |  | D | 0.868 | 1.430 | 0.867 | 1.431 |
> > |  | N | 0.069 | 1.180 | **0.069** | 1.190 |
> > |  | Q | 0.140 | 0.195 | 0.126 | **0.194** |
> >
> > _**Table:** Performance comparison between PINN (P), Physics-Informed Neural Operator (N), DeepONet (D), and PDEDIFF (Q) across different 1D problem settings. **PMSE** is P-MSE and **WD** is the Wasserstein distance (lower is better).  The first two blocks correspond to the **1D Infinite Potential Well**, and **1D GP** corresponds to the **1D Gross–Pitaevskii Equation**._
> >
> > However, we completely agree with the reviewer that PINN (and similar mean estimation methods) should not be a method of choice in cases where multiple solutions are possible as it only learns a mean function. Yet, recent papers that attempt to develop physics-informed diffusion models [1,2] compare to PINNs, and _more importantly do not provide enough evidence that physics-based diffusion models can outperform PINNs_. This is a key message of our paper.
> >
> > We also thank the reviewer for pointing out the GP based PDE papers. Even though GPs model uncertainty and can be used to generate samples from the solution like diffusion models, the inherent Gaussian assumption implies a unimodal function, and hence the goal of GP approaches is still to learn the mean function. As a result, GP based methods cannot handle settings where the PDEs have multiple feasible solutions.
> > Still, we ran a small set of experiments on Gaussian Processes and added the result to section A.7 in the Appendix of the revised paper. We hope this address the weaknesses above and that the reviewer can consider revising their score. We are happy to clarify any further concerns.
> >
> > [1]: Jacobsen, Christian, Yilin Zhuang, and Karthik Duraisamy. "Cocogen: Physically consistent and conditioned score-based generative models for forward and inverse problems." SIAM Journal on Scientific Computing 47.2 (2025): C399-C425.
> >
> > [2]: Huang, Jiahe, et al. "DiffusionPDE: Generative PDE-solving under partial observation." Advances in Neural Information Processing Systems 37 (2024): 130291-130323.

---

### Official Review · Reviewer_6mYT · 2025-10-21

**Soundness:** 2
**Presentation:** 3
**Contribution:** 1
**Rating:** 2
**Confidence:** 4

**Summary:**

This paper proposes a scheme for recovering distributions over solution fields of PDEs using physics informed diffusion models, Specifically they train a DDPM with a PINN style autodiff loss. The method is tested on a few benchmarks looking at eigen decompositions, Schrodinger equations, and helmoltz problems.

**Strengths:**

- The paper is written clearly.

- The proposed method shows advantages over PINNs, however this is not the gold standard in computation of these problems.

- The appendix takes care is carefully explaining the experiments therein.

**Weaknesses:**

- There exist standard methodology for computing the solution to the mentioned problems, such as finite elements with eigen decompositions. Why are these not compared to? It is difficult to assess the usefulness of the proposed methodology if there is not shown to be a concrete advantage of this method against classical schemes.

- I suspect finite difference stencils are more computationally efficient and less memory intensive than using automatic differentiation through the model. However, there seems to be no mention of computational efficiency, runtime evaluation, etc. Why was this not reported?

- Overall the methodology is not very well motivated, as existing tools exist to carry out these computations, and limited testing is presented.

- If the advantage of the method is to blend data and physics, this should much more clearly be stated, explained, and explored with numerical experiments.

- Overall the novelty of the proposed method is quite limited, it mainly encompasses combining an autodiff residual loss with a diffusion model.

**Questions:**

- Can you please explain what the term "modality recall" means? (page 2 contribution 4)

- If one were to use this method, how would you choose the c_res tradeoff parameter in practice? This hyperparameter selection problem is underexplored.

---

> ### Author Response · Authors · 2025-11-21
>
> We thank all reviewers for their time and constructive feedback on our submission. We are encouraged by the positive remarks that recognize our motivation to address multi-modal PDE modeling, which has been a long standing limitation of physics-informed systems. We are glad that the reviewers found the experimental results motivating and recognized our method’s ability to generate diverse, physically consistent solutions. We also appreciate that the reviewers found our paper understandable with sufficient explanations of the experimental setups. We carefully address all concerns below and also pose clarifying questions to better understand how we could improve the scope, evaluation, and clarity of our work in future versions.
>
> **Comparison against numerical methods.** Eigen-solvers require explicit discretization of the differential operator which involves calculating the mesh, domain, stencils and boundary conditions. Whereas, PDEDIFF reduces these bottlenecks and learns $P(Y | X)$, where $Y$ is the solution field for a given spatial input $X$. In our preliminary experiment to compare an eigen-solver (SciPy) with PDEDIFF, we ran an experiment to solve for the 2D Infinite Potential Well. The time taken to solve for 4 eigenstates took approximately $5708$ seconds on a $1000 \\times 1000$ grid whereas after training PDEDIFF, it only requires approximately $3126$ seconds to sample all those $10^6$ points. Generating $10^6$ datapoints using PDEDIFF is $1.8$ times faster as compared to using an eigen-solver.
>
> The advantage of using a diffusion sampler is when we want to sample multiple times for different regions in the domain. In the case of an eigensolver, if we want to find out the value at a new coordinate that is not in the mesh, then we would either have to interpolate it to a grid or re-solve the new mesh with this new coordinate. Whereas, with PDEDIFF, one can sample the required coordinate directly.
>
> **Clarification about modality recall.** "Modality recall" in our context means how well the model generates distinct physical modes such as different eigenstates. A low modality recall would mean that the model collapses to a single mode which is a failure case we discuss for other baseline methods.
>
> **Hyperparameter selection.** As discussed in our Ablation Study in Appendix A.6, we have performed multiple experiments with different $c\_{res}$ values ranging from $0.001$ to $10$. As per our findings, we have discovered that for our method, adding a small value of $c\_{res}$ such as $0.001$ or $0.005$ performs better as compared to using a larger $c\_{res}$ value which would cause the model to collapse to a trivial solution or any of the modes.

---

> ### Comment · Reviewer_6mYT · 2025-11-24
> **response to rebuttal**
>
> Thank you for your answers and comments. Thank you for pointing out Appendix 6.
>
> Has the experiment solving the eigen problem on a 1000 x 1000 grid been added to the paper?  It seems that FEM is likely to perform extremely well on a much more reasonable grid size, such as 50 x 50 or 100 x 100, particularly for the Helmholtz problem you have in figure 5. Furthermore, FEM does not only provide solutions on the node values of a grid, the basis functions themselves define the solution on the entire domain.
>
> Without adding a thorough and comprehensive study of the benefit of your method against the gold-standard classical baseline for such a problem, it is not possible to know if the method you propose is indeed advantageous.

---

> > ### Author Response · Authors · 2025-11-27
> >
> > We will add the result mentioned in the previous post in our paper.
> >
> > We agree that in a problem setting where the physical system is defined using a system of PDEs governing a physical system has a unique solution for fixed known parameters, properly defined boundary conditions and initial conditions with perfect mesh constructions, FEM outperforms all possible methods in terms of speed and accuracy. Additionally, FEM returns the entire basis functions rather than samples from the domain unlike PDEDIFF. However, the motivation and contribution of PDEDIFF is not to replace traditional solvers for well-posed conditions, but to address situations where these methods might struggle.
> >
> > As discussed in Section 1, PDEDIFF is designed to recover solutions from sparse and irregular samples for a system that could inherently have multiple solutions based on the observational data. In these settings, FEMs face a number of challenges. First, FEM cannot reconstruct the solution when random samples are given without using complex inverse problem solvers. Second, with unknown parameters or boundary/ initial conditions, or observational data with a mix of unknown parameter values or boundary/initial conditions, FEMs require solving for different parameter values to see which could lead to convergence. As a result, they can only handle small number of discrete parameter values or boundary/initial conditions, rather than an entire set or non-overlapping sets of parameter ranges. On the other hand, instead of solving for the solution on the entire domain, PDEDIFF can generate samples for different conditions (unknown parameter ranges, boundary or initial conditions) based on observations, in these scenarios. We hope the reviewer will agree with this perspective and consider revising their score. We are happy to clarify any further concerns.

---

### Official Review · Reviewer_vubo · 2025-10-31

**Soundness:** 4
**Presentation:** 4
**Contribution:** 3
**Rating:** 8
**Confidence:** 5

**Summary:**

The authors address a problem faced by PINN approaches: when a partial differential equation (PDE) problem has multiple feasible solutions (multimodality), typical neural network approximators converge toward the mean and "smear" the modes. PDEDIFF is a conditional diffusion model that learns the distribution P(Y∣X) over solution fields and incorporates physical constraints (the equation and boundary conditions) directly into the diffusion target via a residual penalty calculated by automatic differentiation on an arbitrary set of points (mesh-free). Unlike PINN, which minimizes the residual around E[Y∣X] and loses modality, PDEDIFF generates diverse, physically consistent solution samples.

**Strengths:**

1. Clear experiment details. I fully understand the details
2. The multi-modality is very hard goal for achievent in PINNs. And that work gives us a nice vector
3. Good concept, nice toy examples

**Weaknesses:**

1. Not enough baselines, as SPINN, DeepONet, Separable FNO etc.
2. Not enough different equations, like reaction-diffusion and heat equation.
3. I didn't see the experiments with high frequency equations

**Questions:**

Thanks for your paper! I have several questions.
1. Did you check experiments with normalization of coords before learning? It can boost your quality
2. If to be honest, need comparison with DeepONet and SPINN with free parameters.
3. Can you compare your approach on Klein Gordon 3d in SPINN?

---

> ### Author Response · Authors · 2025-11-21
>
> We thank all reviewers for their time and constructive feedback on our submission. We are encouraged by the positive remarks that recognize our motivation to address multi-modal PDE modeling, which has been a long standing limitation of physics-informed systems. We are glad that the reviewers found the experimental results motivating and recognized our method’s ability to generate diverse, physically consistent solutions. We also appreciate that the reviewers found our paper understandable with sufficient explanations of the experimental setups. We carefully address all concerns below and also pose clarifying questions to better understand how we could improve the scope, evaluation, and clarity of our work in future versions.
>
> **Additional results.** We have conducted experiments to compare our method with FNO and DeepONet, both in their Vanilla version and in Physics-Informed versions. To extend our work, we have also conducted an experiment to compare our method on Burgers Equation which covers another non-linear PDE with periodic boundary conditions.
> | Setting | **P**        | **Q**        |
> |---------|----------------------------------|----------------------------------|
> | 0       |$0.0516 $              |$0.0149 $             |
> | 0.001   | $0.0685$             | $0.0148$              |
>
> _**Table:** Preliminary performance comparison between PINN (P) and PDEDIFF (Q) for Burgers' Equation on a specific trajectory with two different viscosities. Values are P-MSE (lower is better)._
>
>
> | Setting | Model | **Vanilla PMSE** | **Vanilla WD** | **Physics-Informed PMSE** | **Physics-Informed WD** |
> |--------|--------|------------------|----------------|----------------------------|--------------------------|
> | **1D (2 states)** | P | 0.822 | 0.539 | 1.052 | 0.344 |
> |  | D | 0.932 | 0.157 | 0.935 | 0.157 |
> |  | N | 0.809 | 0.544 | 0.960 | 0.446 |
> |  | Q | 0.017 | 0.086 | **0.015** | **0.086** |
> | **1D (3 states)** | P | 0.371 | 0.737 | 0.460 | 0.442 |
> |  | D | 0.438 | 0.114 | 0.437 | 0.114 |
> |  | N | 0.333 | 0.776 | 0.445 | 0.470 |
> |  | Q | 0.022 | 0.055 | **0.012** | **0.050** |
> | **1D GP (3 states)** | P | 1.550 | 2.036 | 1.291 | 2.418 |
> |  | D | 0.868 | 1.430 | 0.867 | 1.431 |
> |  | N | 0.069 | 1.180 | **0.069** | 1.190 |
> |  | Q | 0.140 | 0.195 | 0.126 | **0.194** |
>
> _**Table:** Performance comparison between PINN (P), Physics-Informed Neural Operator (N), DeepONet (D), and PDEDIFF (Q) across different 1D problem settings. **PMSE** is P-MSE and **WD** is the Wasserstein distance (lower is better).  The first two blocks correspond to the **1D Infinite Potential Well**, and **1D GP** corresponds to the **1D Gross–Pitaevskii Equation**._
>
> In the 1D Gross Pitaevskii experiment, PiNO achieved a very low PMSE, because it only learn one of the states instead of all states. This reflected in the high WD metric. We will also update the experiments' detail and the full ablation study in the Appendix section in our revision of the paper.
>
> **Coordinate normalization.** We confirm that we did not normalize coordinates. Since the PDE residuals (derivatives) are scale-dependent. If we normalize our coordinates then we would have to de-normalize them while calculating the residual loss, this would add additional complexity to the autograd implementation for the residual calculation. Hence, we decided to not normalize our coordinates.
>
> **High frequency equations.** For the high frequency equations, we have discussed experiment results on 2D Infinite Potential Well with 4 eigen-states in Table 2 of the main section of the paper.
>
> **Questions to Reviewer:** Could you please suggest if there are any specific high frequency equations that we should experiment on? Also, we found a few variations of SPINN and separable FNO, can you please specify the citations that you were referring to?

---

> > ### Author Response · Authors · 2025-11-22
> >
> > **Additional results (2D settings)**
> >
> > | Setting | Model | **Vanilla PMSE** | **Vanilla WD** | **Physics-Informed PMSE** | **Physics-Informed WD** |
> > |--------|--------|------------------|----------------|----------------------------|--------------------------|
> > | **2D (3 non-degenerate states)** | P | 1.253 | 0.378 | 1.239 | 0.340 |
> > |  | D | 7.327 | 0.411 | 7.326 | 0.410 |
> > |  | N | 1.254 | 0.297 | 1.398 | 0.502 |
> > |  | Q | 1.222 | 0.169 | **1.194** | **0.156** |
> > | **2D (4 degenerate states)** | P | 1.250 | 0.482 | 1.243 | 0.477 |
> > |  | D | 7.661 | 0.485 | 7.659 | 0.485 |
> > |  | N | 1.332 | 0.299 | **0.933** | 0.513 |
> > |  | Q | 1.309 | 0.150 | 1.338 | **0.124** |
> > | **2D (4 non-degenerate states)** | P | 1.133 | 0.597 | 1.123 | 0.623 |
> > |  | D | 22.34 | 0.513 | 22.34 | 0.513 |
> > |  | N | 1.198 | 0.344 | 1.257 | 0.587 |
> > |  | Q | 1.100 | 0.222 | **1.091** | **0.205** |
> > | **Helmholtz** | P | 0.800 | 0.107 | 0.746 | 0.108 |
> > |  | D | 21.35 | 0.168 | 21.35 | 0.168 |
> > |  | N | 0.753 | 0.111 | 1.000 | 0.106 |
> > |  | Q | 0.253 | 0.092 | **0.227** | **0.091** |
> >
> > **Table:** Performance comparison between PINN (P), Physics-Informed Neural Operator (N), DeepONet (D), and PDEDIFF (Q) across different 2D problem settings. PMSE is P-MSE and WD is the Wasserstein distance (lower is better). The first three blocks correspond to the 2D Infinite Potential Well, and Helmholtz corresponds to the non-homogeneous Helmholtz Equation. **PMSE** is P-MSE and **WD** is the Wasserstein metric (lower is better).
> >
> > Details of the PDE parameters are provided in the Appendix section of our paper.

---

### Official Review · Reviewer_iF3t · 2025-11-01

**Soundness:** 3
**Presentation:** 2
**Contribution:** 2
**Rating:** 4
**Confidence:** 3

**Summary:**

Authors propose PDEDIFF, a mesh-free physics-informed conditional diffusion framework, to address the key limitation of existing PDE solvers: most methods (e.g., PINNs) collapse multi-modal PDE solutions to conditional means, while standard diffusion models lack explicit physical constraints. The core innovation of PDEDIFF lies in two aspects: first, it learns joint distributions over PDE solution fields and key parameters/eigenvalues, enabling sampling of diverse physically consistent modes; second, it uses automatic differentiation (instead of grid-dependent finite differences) to compute PDE residuals and boundary conditions, supporting sparse, irregular input samples without re-designing solvers for different PDEs. Experiments validate PDEDIFF on 1D/2D linear  and nonlinear PDEs, showing it outperforms PINNs in capturing multi-modal solutions via metrics like P-MSE and Wasserstein-1 distance. Overall, PDEDIFF aims to serve as a generative surrogate for uncertainty-aware multi-modal PDE modeling in low-to-moderate dimensions.

**Strengths:**

- The paper has a clear motivation to solve multi-modal PDE modeling, a long-standing gap of PINNs. Unlike PINNs that regress to conditional means and blur distinct physical solutions (e.g., quantum eigenstates), PDEDIFF explicitly learns the joint distribution of
u(x) and λ, which allows it to sample diverse, physically valid modes—this fills a critical need in physics-informed learning for systems with multiple solutions.
- PDEDIFF’s mesh-free design with automatic differentiation enhances practical flexibility. By replacing finite differences with automatic differentiation for residual calculation, it supports sparse and irregular input samples.

**Weaknesses:**

- The background introduction and problem definition of $\lambda$ (eigenvalue/parameter) are insufficient. The paper treats $\lambda$ as a core component of the joint distribution but fails to clarify $\lambda$ s role across different PDE types. For example, as $\lambda$  is an eigenvalue for Schrödinger equations, and a wave number for Helmholtz equations, what's the position and meaning of $\lambda$ for PDEs without the eigenfunction-value form? In such case, will $\lambda$ still explicitly link to multi-modality, and is still reasonable to generate $\lambda$? More clarification and discussion are encouraged.


- The experiment could be enhanced. The current setting lacks comparisons with state-of-the-art diffusion-based PDE methods like CocoGen[1] and DiffusPDE[2]. Current baselines only include vanila PINNs, which is not convincing enough. Besides, as the paper claimed that the method can be expanded to non-zero boundary condition, it better to demostrate it in the experiment.


- As the work set the mesh-free encoder as a selling point, lack of discussion on existing literature [3,4,5] with similar design limit the novelty of proposed work.

Ref:

[1]: Jacobsen, Christian, Yilin Zhuang, and Karthik Duraisamy. "Cocogen: Physically consistent and conditioned score-based generative models for forward and inverse problems." SIAM Journal on Scientific Computing 47.2 (2025): C399-C425.

[2]: Huang, Jiahe, et al. "DiffusionPDE: Generative PDE-solving under partial observation." Advances in Neural Information Processing Systems 37 (2024): 130291-130323.

[3]: Du, Pan, et al. "Conditional neural field latent diffusion model for generating spatiotemporal turbulence." Nature Communications 15.1 (2024): 10416.

[4]: Chen, Panqi, et al. "Generating Full-field Evolution of Physical Dynamics from Irregular Sparse Observations." arXiv preprint arXiv:2505.09284 (2025).

[5]: Wu, Haixu, et al. "Transolver: A fast transformer solver for pdes on general geometries." arXiv preprint arXiv:2402.02366 (2024).

**Questions:**

Do you have plans to validate PDEDIFF on higher-dimensional PDEs (e.g., 3D Schrödinger equations) and non-Dirichlet boundaries (e.g., Neumann conditions)? If so, please outline technical challenges (e.g., memory for high-dimensional ) and preliminary results

---

> ### Author Response · Authors · 2025-11-21
>
> We thank all reviewers for their time and constructive feedback on our submission. We are encouraged by the positive remarks that recognize our motivation to address multi-modal PDE modeling, which has been a long standing limitation of physics-informed systems. We are glad that the reviewers found the experimental results motivating and recognized our method’s ability to generate diverse, physically consistent solutions. We also appreciate that the reviewers found our paper understandable with sufficient explanations of the experimental setups. We carefully address all concerns below and also pose clarifying questions to better understand how we could improve the scope, evaluation, and clarity of our work in future versions.
>
> **$\lambda$ clarification.** We apologize for the unclear notation on $\lambda$ in our paper. The role of $\lambda$ in different PDEs works as a system parameter, for instance, in the case of eigen-PDEs such as the Schrodinger equation, $\lambda$ represents the eigenvalue or the energy of the corresponding wavefunction. In the case of the non-homogeneous Helmholtz equation, $\lambda$ or $k$ is being used to denote the wavenumber that could correspond to different frequencies and waves that have been collected from a physical setup.
>
> In general form of PDE, $\lambda$ will represent the physical system parameters such as conductivity, viscosity, Young's Modulus and many more depending on the physical system and the PDE describing the system. When PDEDIFF perform sampling, it samples the solution field and this system parameter jointly.
>
> **Literature discussion.** Thank you for the suggested literature. We have compared our method with the given state-of-the-art methods and our problem setting is different from theirs in the following ways: Both Cocogen and DiffusionPDE incorporate physics guidance during the sampling stage whereas we include the physics guidance during the training stage itself. The other methods in the mentioned work assume that only a single valid solution exists for the system and are heavier towards being data driven than physics guidance.
>
> **PDEDIFF on non-Dirichlet boundary condition.** We have conducted a preliminary set of experiments from PDEBENCH [1] dataset on Burgers Equation which is a non-linear PDE and follows a periodic boundary condition. The experiments were conducted in the same setup as other equations discussed in the paper.
>
> | Setting | **P**        | **Q**        |
> |---------|----------------------------------|----------------------------------|
> | 0       |$0.0516 \\pm 0.0024$              |$0.0149 \\pm 0.0025$             |
> | 0.001   | $0.0685 \\pm 0.0119$             | $0.0148 \\pm 0.0023$              |
>
> _**Table:** Preliminary performance comparison between PINN (P) and PDEDIFF (Q) for Burgers' Equation on a specific trajectory with two different viscosities. Values are P-MSE (lower is better)._
>
> **Discussion for high dimensional setting.** For a high dimensional setting, we believe that we would have to explore a bigger architecture for CCED to able to represent high dimensional PDEs such as 4D Klein Gordon or 3D Schrodinger Equation. Also, in higher dimensions the eigen-states in 3D Schrodinger Equation have multiple degenerate energy states and learning a high dimensional probability distribution with sparse observations and high degeneracy is a challenging problem.
>
> Reference:
>
> [1] Takamoto, Makoto, et al. "Pdebench: An extensive benchmark for scientific machine learning." Advances in Neural Information Processing Systems 35 (2022): 1596-1611.

---

### Author Response · Authors · 2025-12-03
**Summary of Discussion Period**

We thank all reviewers for their constructive feedback and participation during the discussion period. We are encouraged that the reviewers recognized our motivation to address multi-modal PDE modeling and its role in bridging a long-standing gap in the field. We also appreciate their positive feedback on our experimental details, conceptual explanations, and the novelty of our mesh-free, automatic differentiation-based approach.

**Bridging the Gap** Reviewers (iF3t, vubo) noted that our method successfully fills a critical gap where standard PINNs and operator learners collapse to conditional means, blurring distinct physical solutions. PDEDIFF is explicitly designed to model uncertainty over multiple feasible solutions for a physical system based on sparse observational data.

**Mesh-Free Flexibility** Reviewers (iF3t, A7D5) emphasized that using automatic differentiation instead of finite-difference stencils enables training on sparse, irregular input samples without redesigning PDE-specific solvers or meshes.

**Multi-Modal Modeling** Reviewers (iF3t, vubo) also recognized as a key contribution that PDEDIFF learns the joint distribution over solution fields and system parameters and can sample diverse, physically valid modes instead of regressing to a single average solution.

In addition, based on the valuable suggestions provided, we have updated our paper. The following is a summary of the main additions and how we have addressed the specific concerns raised:

**New Baselines Added** Beyond PINNs, we have compared PDEDIFF against DeepONet, Fourier Neural Operators (FNO), and Physics-Informed Neural Operators (PINO) for 1D Infinite Potential Well, 1D Gross-Pitaevskii, 2D Infinite Potential Well, and Non-Homogeneous Helmholtz Equations under both degenerate and non-degenerate data in Table 1 and Table 2 of the main paper.

**New Non-Linear Equation on non-Dirichlet boundary** Along with 1D Gross-Pitaevskii (Section 5.2 and Section A.5.2), we have extended our experiments to include the 1D  Burgers’ Equation (Section A.5.5), demonstrating PDEDIFF’s capability to handle non-linear dynamics and periodic boundary conditions.

**Gaussian Process Comparison** We have added a comparison with Physics-Informed Gaussian Processes (EPGP kernels), in Section A.6, as suggested by Reviewer A7D5. Our results show that, even with physics-informed kernels, EPGP still cannot effectively learn truly multi-modal PDE solutions for different pairs of solution fields and system parameters.

**Clarifications** We have fixed the typos as suggested by the reviewers and have added additional statements to explain our motivation behind PDEDIFF and its problem setting.

We have actively addressed the specific weaknesses raised by the reviewers through our revisions:

**Lack of Baselines** Our results show that while FNO and DeepONet achieve good P-MSE scores, they also fail to capture the distribution of solutions (Figure 13), resulting in high Wasserstein Distances (Table 2). Furthermore, the Gaussian Process experiments confirmed that Gaussian Process priors (which assume a Gaussian distribution) fundamentally struggle with disjoint solution modes. These comparisons strongly validate the necessity of the diffusion-based approach for this class of problems (Section A.6).

**Limited Equations and Boundary Conditions** We have successfully demonstrated PDEDIFF's applicability on non-linear PDEs such as Gross-Pitaesvksii and Burgers' Equations with non-Dirichlet boundary as well as on non-homogenous PDEs such as Helmholtz equation. The model successfully recovered the solution trajectories under varying system parameters, showing that the framework generalizes across various family of PDEs.

**Extending Related Works** We have expanded the related works section to discuss recent physics-informed diffusion and mesh-free architectures such as CocoGen, DiffusionPDE, turbulence-oriented latent diffusion models, and Transolver, and clarified how PDEDIFF differs. In PDEDIFF, physics guidance is applied during training and not just sampling, the model is mesh-free via autodiff rather than finite differences, and the focus is explicitly on multi-modal PDEs.

**Comparison to Numerical Methods** We have clarified that PDEDIFF is not intended to replace FEM or other classical solvers in well-posed scenarios with fully known parameters, clean boundary/initial conditions, and high-quality meshes, where FEM indeed remains the gold standard. Instead, PDEDIFF targets data-driven scenarios with sparse and irregular observations and unknown or mixed parameter settings, where classical methods require complex inverse formulations and repeated solves. Additionally, our analysis shows that for generating a large ensemble of data points and solutions, PDEDIFF offers a speed advantage over resolving the system numerically for every sample (Sec A.7).

---

> ### Author Response · Authors · 2025-12-03
> **Final Remark**
>
> Throughout the discussion period, we have clarified the unique aspects of PDEDIFF, and the reviewers’ comments have been invaluable in helping us articulate our contributions and motivation more clearly. We hope that our revisions sufficiently address the concerns raised, and we would be grateful if you could kindly consider these updates when assessing any potential adjustments to the reviews and the overall decision.

---

### Meta-Review · Area_Chair_pzbK · 2026-01-06

**Summary:**

Common concerns includes:
1. Almost all reviewers express concern about baselines. For example, Reviewer iF3t points out that the paper hasn't compared to DiffusionPDE and CocoGen, which proposed very similar idea of solving PDEs as sampling from a conditional distribution. Reviewer vubo, despite his/her high rating, concerns about the paper not comparing with enough baseline (SPINN, DeepONet, Separable FNO), not enough PDEs, and not enough high-frequency PDEs. Reviewer ymYT concerns about FEM baseline. Reviewer A7D5 concerns about Gaussian Process baseline.
2. Limited Novelty. Reviewer iF3t points out the similarity between this work and some prior works like DiffusionPDE and CocoGen. Reviewer 6mYT also concerns about novelty.
3. Justification of Computation comparing to baseline methods. Reviewer 6myT points out that the proposed method could be computationally expensive when comparing to classical methods, so it is important to justify such additional computes.

**Reviewer Concerns:**

The authors provide many additional experiment results, including some toy baseline results using FEM on high-resolution images, results from PDEBench, additional non-linear PDE results, and comparisons with additional baselines that could output multiple different states for underspecified PDEs.

Authors also provides in rebuttal clarifications about technical questions as well as a modified PDEs containings changes.

The main concerns left unaddressed includes:
1. How does the method compare to PDE + diffusion baselines empirically? While the authors addressed the difference in the related works, the difference doesn't prevent these baselines from performing the same tasks the proposed method aims to. Yet authors haven't provide empirical evidence showcasing the advantage of proposed method.
2. Justification of additional computes comparing to classic methods like FEM. While the authors provides in the appendix where a high-resolution FEM solvers might takes longer than the proposed diffusion based method, the experiment is preliminary as two methods can be profiled by different hardwares.

**Reviewer Scores:**

Reviewer vubo might likely maintain his/her score. Reviewer A7D5 might to increase the score toward borderline accept if given the time to inspect the GP results. Reviewer 6mYT and iF3t might finds the response only partially satisfied.

---

### Decision · Program_Chairs · 2026-01-26

Reject